# A variational regularization of Abel transform for GPS radio occultation

Tae-Kwon Wee

University Corporation for Atmospheric Research, Boulder, Colorado, USA

*Correspondence to*: Tae-Kwon Wee (wee@ucar.edu)

**Abstract.** In the Global Positioning System (GPS) Radio Occultation (RO) technique, the inverse Abel transform of measured bending angle (Abel inversion) is the standard means of deriving the refractivity. While concise and straightforward to apply, the Abel inversion (AI) accumulates and propagates the measurement error downward. The measurement error propagation is detrimental to the refractivity in lower altitudes. In particular, it builds up negative refractivity bias in the tropical lower troposphere. An alternative to AI is the numerical inversion of the forward Abel transform, which does not incur the integration

of error-possessing measurement and thus precludes the error propagation. The variational regularization (VR) proposed in this study approximates the inversion of the forward Abel transform by an optimization problem in which the regularized solution describes the measurement as closely as possible within the measurement's considered accuracy. The optimization problem is then solved iteratively by means of the adjoint technique. VR is formulated with error covariance matrices, which permit a rigorous incorporation of prior information on measurement error characteristics and the solution's desired behaviour

into the regularization. VR holds the control variable in the measurement space to take advantage of the posterior height determination and to negate the measurement error due to the mismodelling of the refractional radius. The advantages of having the solution and the measurement in the same space are elaborated using a purposely corrupted synthetic sounding with a known true solution. The competency of VR relative to AI is validated with a large number of actual RO soundings. The comparison to nearby radiosonde observations shows that VR attains considerably smaller random and systematic errors

compared to AI. A noteworthy finding is that in the heights and areas that the measurement bias is supposedly small, VR follows AI very closely in the mean refractivity deserting the first guess. In the lowest few kilometers that AI produces large negative refractivity bias, VR reduces the refractivity bias substantially with the aid of the background, which in this study is the operational forecasts of the European Centre for Medium-Range Weather Forecasts (ECMWF). It is concluded based on the results presented in this study that VR offers a definite advantage over AI in the quality of refractivity.

## 1 Introduction

The Abel transform pairs (Abel, 1826) are widely used to reconstruct radially (or spherically) symmetric physical parameters from their line-of-sight (LOS) projections in a variety of disciplines in engineering and science. In the Global Positioning System (GPS) Radio Occultation (RO) technique, inverse Abel transform (often-called Abel inversion) of the bending angle in particular has become a cornerstone, serving as the standard means of deriving the refractivity. Knowledge of the refractivity

structure in the atmosphere is important for numerous applications relevant to weather and climate. The LOS projection in RO corresponds to the phase or ray's bending angle. Hence, these are referred to as RO measurements hereafter, unless otherwise mentioned.

The Abel Inversion (AI hereafter) is mathematically exact, meaning that AI is supposed to facilitate a unique and perfect reconstruction of the symmetric media, given the measurement of infinite accuracy and resolution. However, measurements in real life are noisy and available only at a discrete set of data points. Some of the previous studies in the literature are focused on dealing with the data resolution issue and others are on reducing adverse effect of the measurement noise. Existing methods attempt to improve the accuracy of discrete Abel transforms by employing higher-order numerical schemes (e.g., Kolhe and Agrawa, 2009), polynomial interpolation or fitting methods (e.g., Deutsch and Beniaminy, 1983), the Fourier transform (Kalal and Nugent, 1988), and the Fourier-Hankel transform (Ma, 2011). The relative significance of the two issues (i.e., data resolution and measurement noise) depends on the medium (problem) of interest and the observing system used to sample the LOS projection, and so does the performance of these methods.

In case of RO, the data resolution might be high enough to not cause significant discretization error. (Here, the term "discretization error" means the error incurred from approximating the analytical integral transform using a finite number of discrete data points.) Nevertheless, RO measurements are subject to non-negligible errors arising from diverse sources (e.g., Kursinski et al., 1997; Hajj et al., 2002; Steiner and Kirchengast, 2005) to which AI is sensitive. In addition, the premise of AI (i.e., spherically symmetric atmosphere) is never strictly fulfilled. Gorbunov et al. (2015) claim that strong horizontal refractivity gradients can cause the bending angle to be a multivalued function of the Impact Parameter (IP). This relates to the fact that the IP is not conserved along a ray path in the horizontally inhomogeneous atmosphere (Healy, 2001). The change of IP along the ray path can be as big as 80 m, which corresponds to ~4% deviation in refractivity (Wee et al., 2010). This in turn indicates that it is generally impossible to assign a specific IP to a single ray or to a unique value of the bending angle. In other words, an IP can be associated with multiple values of the bending angle. This causes a highly scattered distribution of "raw" (unsmoothed) bending angles in the IP coordinate. The bending angle that is not a well-defined function of IP (i.e., multi-valued or greatly dispersed) accompanies a large data uncertainty, which in turn propagates into the refractivity. Thus, horizontal inhomogeneity (either large-scale gradient or small-scale fluctuations) of the refractivity causes additional measurement error, which is largely random and greater in the lower atmosphere.

RO measurements contain systematic error as well. For example, the phase and bending angle are often undermeasured in the lower troposphere. Some of the potential causes are imperfect signal tracking (Sokolovskiy et al., 2010), critical refraction (Sokolovskiy, 2003; Ao et al., 2003; Xie et al., 2006), and small-scale refractivity fluctuations (Gorbunov et al., 2015). The critical refraction is probably the most well understood among the causes, thanks to ingenious previous studies. When the critical refraction occurs, the bending angle is unbounded. Specifically, the bending angle goes to infinity at the height of the critical refraction [e.g., see Fig. 5b of Sokolovskiy (2003)]. In RO, the refractivity is obtained from integrating the bending angle vertically, where the vertical weighting is given by the Abel kernel. Since measured bending angles are finite in magnitude, the Abel integral results in a negatively biased refractivity below the top of the ducting layer. (Throughout this

paper and for the sake of convenience, the term "bias" is used interchangeably with systematic error when the reference for the bias is omitted.) Even under conditions of subcritical refraction, RO bending angles tend to be negatively biased. A brief explanation can be as follows. Highly bending rays arrive at the receiver when the receiver is far behind the Earth's limb. The arrivals are recorded at the trailing (leading) epochs of sinking (rising) occultation events. Being weak and noisy, the signals received during the epochs have a higher chance of not being used for bending angle estimation. The exclusion of those measurement pieces leads to the loss of information on highly refracting rays and in turn to a negatively biased bending angle. In that case, RO bending angle is inclined to have local peaks weaker than they are supposed to be. Reader are referred to Sokolovskiy et al. (2010) for more details. AI integrates the negative bias of the bending angle and turns it into a negative refractivity bias.

What is more concerning in the use of AI is not the measurement error itself but its vertical propagation. AI accumulates and propagates the measurement error in vertical direction. Consequently, a single corrupted piece of the measurement affects not just the location of the particular datum but a wide area that the Abel kernel dictates, and thereby deteriorates the derived refractivity even in the region that received RO signals are clean. Therefore, it is crucially important to moderate the unwanted effect of measurement error. When AI is used, one can take two straightforward approaches: 1) employ noise-resistant numerical methods for the discrete AI or 2) apply a data smoothing to the measurement in advance of the AI. In regard to the former, it is difficult to ascertain which of the numerical methods performs the best. For instance, a method that is more sensitive to noise also shows a higher inversion accuracy for data without noise (Ma, 2011), because high-frequency signal components (i.e., legitimate small-scale structures in the measurement caused by the atmosphere) are seldom distinguishable from noise. The latter has the same difficulty because the data smoothing can be either insufficient or excessive. Hence, neither of those approaches can provide a decisive answer to the issue. Moreover, these approaches are hardly effective for reducing systematic error and for restraining the error propagation. This necessitates alternative approaches.

The critical refraction is an excellent example for understanding the effect of error propagation. It also gives some insights into potential remedies. The use of AI under critical refraction conditions results in a negatively biased refractivity, even with unbiased bending angles (Gorbunov et al., 2015). Indeed, the bending angle bias due to the critical refraction could be confined in the close vicinity of the ducting layer, which is usually very thin. A bending angle sounding unbiased elsewhere can then be considered to be "virtually" unbiased. No matter how shallow the ducting layer is, however, AI propagates the bending angle bias in the layer downward and yields a refractivity sounding that is negatively biased below the ducting layer. A noteworthy point here is that in the reverse modelling perspective, the biased refractivity is not the unique solution attainable from the "unbiased" bending angle. As a matter of fact, multiple refractivity soundings can replicate the bending angle very closely when mapped into the bending angle through the forward Abel transform (FAT) that is the exact mathematical inverse of AI. For instance, any refractivity soundings that are identical with the AI-produced refractivity above the ducting layer but are different by a constant offset below can reproduce the bending angle nearly perfectly except for within the layer of the singularity. That is because the vertical gradient of the refractivity is of prime importance in FAT, but the refractivity itself is not. Hypothetically, one of those soundings is the perfect bias-free refractivity sounding that AI could have attained, if the

critical refraction did not occur. This suggests that the agreement between measured and modelled bending angles can be a clue to finding a refractivity sounding that is less biased than what AI provides. An important implication here is that the solution search does not incur propagation of the bending angle bias, which is unavoidable with AI. The prospect can be extended to conditions of subcritical refraction, as far as the measurement error is considerable and its downward propagation

is concerning. The question is how the "best" refractivity sounding can be chosen among an infinite number of candidates that can replicate approximately well a given bending angle sounding. A sensible metric in the maximum likelihood framework might be able to assist the selection. Another question is how to choose a reasonably small number of good candidates that are around the unknown "true" solution to begin with, given that the cost of FAT can be prohibitive when applied to a large number of soundings. Instead of choosing arbitrary candidates, an approximate inverse solution of FAT can be sought

numerically. The approximate solution can then be perturbed to provide the candidates or it can be successively corrected to approach the true solution. Based on this idea is the regularization approach.

Regularization methods solve the inverse problem numerically. In our setting, the methods seek the inverse solution of FAT, which is the refractivity sounding that reproduces a given bending angle sounding. For measurements without noise, an ideal numerical inversion (note that it differs from numerical implementations of the analytical AI) yields the same solution with

AI. In the presence of measurement noise, however, numerical inversions for problems of the kind of FAT are known to be ill-posed and incapable of providing stable solutions. For instance, different realizations of measurement noise albeit small in magnitude could lead to entirely different solutions. It is thus difficult to obtain useful solutions by applying straightforward algebraic inverse operators to the discrete FAT with noisy measurements. To tackle the ill-posedness, regularization methods enforce regularity on the computed solution, while allowing the solution to deviate from the approximately accurate

measurements. The methods search for the solution by minimizing a joint function that consists of the data fidelity term (which gauges the discrepancy between measurement and its model counterpart) and the penalty term (also called regularization term as it regularizes the solution). These terms used for practical applications are usually simple in the form. For instance, the Tikhonov regularization (Tikhonov, 1963) expects a desirable solution to be spatially smooth and the penalty term widely used for the method is the sum of squared gradients of the solution. Sofieva et al. (2004) claim that the smoothness constraint

improves significantly the quality of their ozone retrievals. Yet, they report the difficulty of using the constraint optimally in the Tikhonov regularization, which is due to the fact that the smoothness measure is sensitive to the data resolution. While the smoothness constraint may work acceptably for some general problems, it alone would be insufficient for more demanding applications since it does not hold any specific information on the desired solution. Generally speaking, the regularization must be customized for individual applications in order to be maximally effective. For instance, different observing systems

differ in the measurement characteristics and the solution's desired behaviour. Thus, the specifics of an optimal regularization differ from one problem to another and those for the Abel transform in RO are underexplored. Another practical difficulty is the fact that regularization methods can be expensive when applied to RO. That is because the limb-viewing geometry offers high-resolution measurements and one typically attempts to retrieve a profile of refractive index in as much detail as the data permit.

This study proposes and studies a variational regularization (VR hereafter) for the Abel transform in GPS RO. The purpose is to improve the quality of RO refractivity by enhancing RO measurement with the aid of prior information. The focus is on the lower troposphere where AI is hampered by larger measurement uncertainty, especially by a considerable negative measurement bias. This study aims at tackling the root cause that degraded the AI-produced refractivity in the first place, which is hypothesized as uninhibited vertical propagation of the measurement error. The observation used in VR is RO bending angle and the state (and control) variable is the refractivity as a function of the refractional radius (RR; the Earth's radius multiplied by the refractive index). As will be explained later, it is essential to define the state variable as a function of the RR. The aptness of data fidelity and penalty terms is vital for a successful regularization. The two terms exploit the prior information on the characteristics of the measurement and solution, which can be either statistical or empirical. In this study, the two terms are formulated with error covariance matrices (ECMs), which succinctly describe the statistical error characteristics. Needless to say, these ECMs must be factual for the formulation to be effective. Meanwhile, regularization methods need a first guess to start off. The first guess used in this study is short-term operational forecasts of the European Centre for Medium-Range Weather Forecasts (ECMWF). Modern-day numerical weather forecasts are comparatively accurate and routinely available. More importantly, a number of well-established methods such as those based on innovation (observation minus forecast) statistics are available for the forecasts, offering reliable error estimates and supporting the construction of ECMs. These methods make available the error estimate of observations as well as that of the forecasts. Hence, short-term forecasts are a compelling source of the first guess. The computational cost is an impediment to the practical use of regularization methods. The proposed method solves the underlying inverse problem iteratively by means of the adjoint method, which is a very efficient way of calculating the gradients of the cost function with respect to all control parameters at once. Accordingly, the variational technique (i.e., gradient-based optimization method) reduces otherwise excessive computational expenses and is thus indispensable for VR. While the Tikhonov regularization is devised based upon the variational principle, it does not necessarily employ the variational technique to be applied to practical problems. It must be mentioned that the adjective used to describe the proposed method (i.e., variational) is meant for the variational procedure, rather than for the variational principle. Therefore, VR purports to indicate a regularization that makes use of the variational technique. Using the ECMs derived based on error statistics and relying on the iterative minimization procedure, the proposed method can also be described as an iterative, statistical regularization.

The variational method has been applied to a variety of problems in diverse areas. The most popular use of the method in GPS RO is data assimilation (DA), but the method is also applied to other estimation problems. An example is the variational combination of dual-frequency RO measurements (Wee and Kuo, 2014), which attempts to optimally separate ionospheric and atmospheric effects. The focus of DA is on the maximal utilization of all available observations, where the forecast model is used as sophisticated physical and dynamical constraints. While assimilating RO data, it is important to take into account contemporaneous observations including those made available by other observing systems. For instance, an overweighting given to RO data leads to an underutilization of other observation types, which in turn results in a suboptimal data assimilation. In addition, RO data assimilation is constrained by the geometry of the model grid, which is not the case for VR.

The proposed method is more akin to one-dimensional variational (1D-Var) retrieval methods (e.g., Healy and Eyre, 2000; Palmer et al., 2000; Palmer and Barnett, 2001; Von Engeln et al., 2003). The Data Analysis and Archive Centre (CDAAC) at the University Corporation for Atmospheric Research (UCAR) has also been using a 1D-Var, developed by the author, for the last 15 years. The key differences between VR and 1D-Var lie in the problem dealt with and the purpose, which deserve further

explanations. The main purpose of RO 1D-Var is to challenge an underdetermined problem in which three variables (i.e., temperature, moisture, and pressure) must be retrieved out of a single observed parameter. Hence, 1D-Var seeks the optimal combination of the state variables utilizing the physical relationship among them and possibly the multi-variate character of the background ECM. Another important difference between 1D-Var and VR is that the state variables of 1D-Var are given as functions of the height (or pressure), whereas the sole state variable of VR, refractivity, is defined in the RR coordinate. In

order to model the bending angle, 1D-Var must simulate the refractivity with the state variables prior to using a discrete FAT. Next, the simulated refractivity is used to compute RR, which defines the location of measured bending angles in the model space. It is worth mentioning that when associated with measured bending angles, the RR represents the one to the ray's tangent point and is the same with the IP in magnitude. It means that the RR in relation to the bending angle is the model counterpart of an IP. It must be pointed out that there are an infinite number of different combinations of temperature, pressure,

and moisture that lead to an identical refractivity. Likewise, a countless number of dissimilar combinations of the refractivity and radius result in the same RR. In the reverse modelling sense, these ambiguities, absent in VR, introduce extra uncertainty to the 1D-Var retrieval. Moreover, the state variables of the 1D-Var usually contain significant errors and so is the modelled refractivity. Again, the refractivity error is carried forward into RR. The problem here is that the erroneous modelled RR is used by 1D-Var as the coordinate to locate the measurement and is thus assumed to be correct by definition. As a result, 1D-

Var cannot perceive a measured sounding of the bending angle as it is. Instead, only distorted (or fuzzy) images of the original sounding are visible to 1D-Var. That is to say, the measurement is always incorrect in the model's perspective, unless the modeled refractivity is perfect. That introduces additional uncertainty to the measurement, although the measurement is not to blame. Eventually, the RR error depreciates the value of measured bending angles.

An important implication here is that the 1D-Var (and DA as well) provided with error-free measurements is unable to recover

the refractivity as a function of the height perfectly, unless the state variables are initially perfect. In VR, on the other hand, the soundings of bending angle and refractivity have one-to-one correspondence in the RR space through a FAT, unless the critical refraction occurs. Given the perfect bending angle, it is thus possible for VR to at least hypothetically reconstruct the perfect refractivity. Therefore, VR is well poised to estimate the refractivity with measured bending angles. In addition, the problem of retrieving temperature, moisture, and pressure can be dealt with separately once the optimal refractivity is made

available and is thus put aside in this study. Doing so eliminates the above-mentioned ambiguities and greatly simplifies the estimation of refractivity with VR.

This study is motivated by the conception that the vertical propagation of measurement errors in AI is detrimental to the derived refractivity and VR is deliberated as a potential remedy to the problem. While VR propagates the model error through FAT, it is not a major concern since VR iteratively corrects the model state to approach the true state with the aid of the

measurement. The focal point of this study will be examining whether and how the deprivation of the measurement error propagation with VR is beneficial to the quality of RO refractivity. The remainder of this paper is organized as follows. Section 2 describes the methods relevant to this study, which include the Abel inversion, the Tikhonov regularization, and the proposed variational regularization. Section 3 compares the proposed method against the Abel inversion with a synthetic RO sounding. Section 4 presents real data tests, along with the verification with two radiosonde data sets. Section 5 offers concluding remarks.

## 2 Method

### 2.1 Abel inversion

The total transpired phase path of a radio wave that propagates through the atmosphere between a transmitter and a receiver can be described by an integral equation (e.g., Wallio and Grossi, 1972):

$$\Phi(x) = 2 \int_x^\infty \frac{n(r)r\,dr}{\sqrt{r^2 - x^2}}, \tag{1}$$

where $\Phi$ is the phase path; $r$ is the radius from the Earth's curvature center; and, $n$ is the refractive index that relates to the refractivity $N = 10^6(1 - n)$. A change of variable can show that Eq. (1) is equivalent to standard form of the Abel transform (Bracewell, 1978). A geometrical interpretation of Eq. (1) is that the Abel transform of $n(r)$, right-hand side (RHS) term, is the projection of $n(r)$ onto the space of the traverse coordinate, $x$, which indicates the nearest approach of the line of sight to the Earth's curvature centre. Being an Abel transform, the analytical inverse of Eq. (1) also exists (Ahmad and Tyler, 1998):

$$n(r) = -\frac{1}{\pi} \int_r^\infty \frac{d\Phi}{dx} \frac{dx}{\sqrt{x^2 - r^2}}. \tag{2}$$

The inverse Abel transform, Eq. (2), provides a straightforward solution to the reconstruction of the refractive index given measurements of phase path. However, the inverse transform is of limited usefulness for practical applications with real-world data because the derivative in RHS term exacerbates the phase noise or the artefacts introduced by arbitrary noise mitigations. Another limitation of Eqs. (1) and (2) is being valid only for a thin, spherically symmetric atmosphere in which the ray's path can be adequately approximated by the straight line that connects the transmitter and receiver. A variant of Eq. (1) suitable for dense, optically stratified media (Fjeldbo et al., 1971) is:

$$\alpha(a) = -2a \int_{r_o}^\infty \frac{d\ln(n)}{dr'} \frac{dr'}{\sqrt{(nr')^2 - a^2}}, \tag{3}$$

where $\alpha$ is the ray's bending angle, $a$ is the IP, and $r_o$ is the radius to the ray's tangent point. With a change of variable, $x = nr$, Eq. (3) can be rewritten as:

$$\alpha(a) = -2a \int_a^\infty \frac{d\ln(n)}{dx} \frac{dx}{\sqrt{x^2 - a^2}}. \tag{4}$$

The corresponding inverse transform is:

$$\ln(n) = \frac{1}{\pi} \int_a^\infty \frac{\alpha(x)dx}{\sqrt{x^2-a^2}}. \tag{5}$$

Equation (5) has the advantage of not having the derivative in RHS term, in addition to accounting for refracting ray paths. For these reasons, Eq. (5) is the AI commonly used in GPS RO. However, there is a caveat - GPS RO does not measure the bending angle directly. Accordingly, the bending angle must be estimated in some ways in connection with the Doppler shift, which again relates to the phase derivative in either time or frequency domain (e.g., Kursinski et al., 1997; Hajj et al., 2002). Therefore, the AI using Eq. (5) is not entirely free from the derivative operator. Instead, the procedure of retrieving the refractivity from the measured phase, equivalent to Eq. (2), is split into two sequential steps: bending angle estimation and subsequent Abel inversion. The derivative operator is put to use before or within the bending angle estimation, although it does not appear explicitly in Eq. (5). No matter when the derivative is used, it intensifies the effect of measurement noise. A data smoothing might be applied to the bending angle in advance of AI. However, the smoothing degree is very difficult to control. Moreover, data smoothing does not reduce systematic measurement error. As explained in the introduction, the vertical propagation of measurement error due to AI is detrimental to the derived refractivity. An alternative to AI is the regularization approach, described in the following.

**2.2 Tikhonov regularization**

The Tikhonov regularization (TR hereafter) is the most widely used regularization method and is indeed the very method that opened up the concept of regularization. Here, a sketch of TR is provided in the context of GPS RO data processing. The general purpose of TR is to solve ill-posed inverse problems in which the forward operator $H(\mathbf{x}) = \mathbf{y}$ defines a mapping $H: X \rightarrow Y$, between the solution (model) space $X$ and the data (measurement) space $Y$. Here, $\mathbf{x}$ is the state vector consisting of model parameters and $\mathbf{y}$ is the vector of modelled observation. In our setting, the forward operator is Eq. (4), and $\mathbf{x}$ and $\mathbf{y}$ hold the refractive index and bending angle, respectively. The method solves a minimization problem of a real-valued function $J$ such that the minimizer, $\mathbf{x}$ , is a suitable approximate solution:

$$\min_{\mathbf{x} \in X} J(\mathbf{x}) = F(\mathbf{x}) + \xi S(\mathbf{x}), \tag{6}$$

where $F$ is the data fidelity term, $S$ is the penalty (regularization) function, and $\xi$ is the regularization parameter. The data fidelity term $F$ measures the misfit between measured and modelled observations, whereas the penalty term $\xi S$ weighs the degree of irregularity in the solution. The scalar parameter $\xi$ controls the relative contribution of the two terms to $J$. A widely used form of $F$ is the squared $L^2$ norm, $F(\mathbf{x}) = \|H(\mathbf{x}) - \mathbf{y}_o\|^2$, where $\mathbf{y}_o$ contains the measured bending angle. A popular choice for $S$ aimed at noise reduction is a seminorm, $S(\mathbf{x}) = \|\nabla \mathbf{x}\|^2$, where $\nabla$ is the gradient operator. Determining the trade-off between the two terms, a decreasing $\xi$ steers Eq. (6) toward (4). Therefore, the solution of TR approaches AI as $\xi S$ vanishes. A drawback of this penalty term is the difficulty of determining the proper $\xi$ that can achieve the optimal smoothness of the

solution. Another difficulty relates to the fact that the vertical refractivity gradient has a high spatiotemporal variation. For instance, $\nabla n$ near the Earth's surface can be a few orders greater than that in the stratosphere. As such, the particular $S$ of the form shown above is lacking as it does not account for the height dependency. Likewise, the $F$ shown above tacitly assumes that the magnitude of $H(\mathbf{x}) - \mathbf{y}_o$ is unvarying in space and time, which is not the case for the bending angle. While the generic TR comes in handy for applications to general problems, it is desirable to utilize the problem-specific prior information in order to make the regularization more effective. In addition, regularization methods can be costly when applied to RO data. It is thus crucial to reduce the cost to a feasible level. The variational regularization proposed in this study aims at addressing these issues.

## 2.3 Proposed method of variational regularization

2.3.1 Formulation

The proposed regularization searches for the solution, $\mathbf{x}$, by minimizing a cost function defined as follows:

$$J(\mathbf{x}) = \frac{1}{2}(\mathbf{x} - \mathbf{x}_b)^{\mathrm{T}} \mathbf{B}^{-1}(\mathbf{x} - \mathbf{x}_b) + \frac{1}{2}\{\mathbf{y}_o - H(\mathbf{x})\}^{\mathrm{T}} \mathbf{R}^{-1}\{\mathbf{y}_o - H(\mathbf{x})\}, \tag{7}$$

where $\mathbf{x}_b$ is the background, a priori of $\mathbf{x}$; $\mathbf{B}$ and $\mathbf{R}$ are the ECM of $\mathbf{x}_b$ and $\mathbf{y}_o$, respectively; and, superscripts "T" and "-1" indicate transpose and inverse of a matrix, respectively. Conceptually, the method seeks the optimal solution that replicates the observation as closely as possible (compelled by the second RHS term) in the vicinity of a priori state (constrained by the first RHS term). The departure of $\mathbf{x}$ from the background (observation) is determined by the assumed error of $\mathbf{x}_b$ ($\mathbf{y}_o$), represented by $\mathbf{B}$ ($\mathbf{R}$). The background (first RHS) term corresponds to the penalty term of TR in the role and the observation (second RHS) term is equivalent to the data fidelity term. Despite the correspondence, the use of ECMs makes VR advantageous over TR, allowing VR to incorporate the prior information about the data uncertainty of $\mathbf{y}_o$ and $\mathbf{x}_b$ into the regularization. The forward observation operator used in VR is Eq. (4), which models the observation (bending angle) with the state variable (refractivity). Equation (4) states that the modelling of bending angle at an IP needs the vertical structure of refractivity above the IP. In order to model a sounding of bending angle, the refractivity must be available along the connected line of tangent points. Given that the RR to the tangent point equals with the IP, it is fair to say that the state variable resides in the IP space where the observation exists. In other words, what we need to model the bending angle is nothing but the refractivity as a function of the IP. Therefore, it is unnecessary for VR to relate the state variable to the temperature, pressure, and moisture during the minimization. In addition, the reverse (adjoint) modelling as well as the forward modelling does not incur any coordinate transform between RR and height, which is inevitable for data assimilation and 1D-Var. This greatly simplifies the problem of refractivity estimation. The coordinate transform entails two issues of great consequence, as explained in the following.

In RO data assimilation, the location of the state-vector elements is represented in relation to the model's native grids. In order to place measured bending angles in the model space and vice versa, it is thus necessary to relate the IP to

the RR, $x = nr$. Based on the never-perfect model's refractivity, $x$ is not a definite measure of the position. That is, it always contains some error. Because the error-possessing RR is used as the coordinate, however, it is free of error by definition. Consequently, the position of the observation in the model space cannot be determined correctly not because of its own error but because of the model's error in the refractivity. This is known as the Errors-In-Variables (EIV) problem in which the error in the independent (or coordinate) variable causes an apparent error in the dependent variable (observation). The EIV error leads to a suboptimal use of the observation. In VR, on the other hand, the EIV error is attributed to the uncertainty of the background refractivity because the model space coincides with the observation space. In the proposed method, the geometric height of the solution keeps changing implicitly during the iterative minimization of the cost function, whereas the RR assigned to the solution remains fixed. Upon the completion of the minimization, the solution's height $z$ is determined by: $z = r - R_c = xn^{-1} - R_c$, where $R_c$ is the local curvature radius of the Earth. Therefore, the solution's height in VR is undecided until the solution is acquired, which is the same as in AI. This Posterior Height Determination (PHD) reduces a substantial portion of retrieval error when viewed in the height coordinate. More details on the topic are presented in the next section.

In the proposed method, a good background furnishes faster convergence to the solution and assists in attaining the desired global minimum of the cost function. Another factor of particular importance in VR is the adequacy of ECMs. That is, the ECMs must be factual, representing well the error characteristics of the background and observation that are actually used in VR. In that respect, short-term forecast of Numerical Weather Prediction (NWP) models is the best source of the background. In addition to being routinely available and of good quality, the forecasts, when used along with relevant observations, offer rigorous error estimation and realistic modelling of the ECMs. Although a climatology is usable as the background, for instance, it does not precisely describe the atmosphere at the exact moment and location of the observation. Besides, it is not as easy to estimate or define the error of the climatology. In this study, short-term operational forecasts of the ECMWF are used as the background. Additional description of the ECMWF data will be provided later in Sect. 4.1.

### 2.3.2 Error covariance matrices

#### a) Error standard deviation

A diagonal element of ECM is the error variance, the square root of which (error standard deviation) represents the error estimate at the location linked to the element. The error variances are estimated by applying the Hollingsworth-Lönnberg method (Hollingsworth and Lönnberg, 1986) to ~1.5 million closely located (< 3 h and < 300 km) pairs of RO soundings available for a 7-year period (April 2007 – April 2014). This method (HL86 hereafter) is based on the innovation statistics with the assumption that forecast errors are mutually (spatially) correlated, whereas observation errors are uncorrelated with themselves and with the forecast errors. In the following, the usage of HL86 in this study is briefly described. Because the error variances can be estimated independently at individual height levels, let $\mathbf{Y}_o$ be the collection of RO observations in a same impact height ($\equiv a - R_c$). That is, $\mathbf{Y}_o = (y_o^k, k = 1, \dots, m)$, where $y_o^k$ is the observation from $k$-th RO sounding out of total $m$ soundings. Likewise, let $\mathbf{Z}_f = (z_f^1, \dots, z_f^m)$ be the forecast counterpart of $\mathbf{Y}_o$, where $z_f^k = \widetilde{H}(\mathbf{x}_f^k)$, $\widetilde{H}$ being the relevant

observation operator. The innovation ($\mathbf{d} \equiv \mathbf{Y}_o - \mathbf{Z}_o$) variance can then be written as:

$$Var\{\mathbf{d}\} = E\{(\mathbf{d} - E(\mathbf{d})^2\} = E\left\{\left(\boldsymbol{\varepsilon}_o - \boldsymbol{\varepsilon}_f\right)^2\right\} = \sigma_o^2 + \sigma_f^2, \tag{8}$$

where $E(\cdot)$ denotes the statistical expectation (i.e., the mean over the $m$ samples); and, $\sigma_o^2$ and $\sigma_f^2$ are the variance of observation error $\boldsymbol{\varepsilon}_o$ and forecast error $\boldsymbol{\varepsilon}_f$, respectively. The mean difference between $\mathbf{Y}_o$ and $\mathbf{Z}_f$ is subtracted in Eq. (8) and

so it does not contribute to $\sigma_o^2$ and $\sigma_f^2$. The random errors, $\boldsymbol{\varepsilon}_o$ and $\boldsymbol{\varepsilon}_f$, are further assumed to be mutually uncorrelated, $E(\boldsymbol{\varepsilon}_o\boldsymbol{\varepsilon}_f) = 0$.

Meanwhile, the innovation covariance between a pair of observations, $y_o^i$ and $y_o^j$, can be written as:

$$cov\{y_o^i - z_f^i, y_o^j - z_f^j\} = E\{(\varepsilon_o^i - \varepsilon_f^i)(\varepsilon_o^j - \varepsilon_f^j)\} = E(\varepsilon_f^i\varepsilon_f^j) = \rho_{ij} \cdot \sigma_f^2 \cong exp\left(-\frac{d_{ij}^2}{2L^2}\right) \cdot \sigma_f^2, \tag{9}$$

where $d_{ij}$ is the horizontal distance between $y_o^i$ and $y_o^j$; $\rho_{ij}$ is the spatial correlation between $\varepsilon_f^i$ and $\varepsilon_f^j$; and, $L$ is the error

correlation length scale. Note that only forecast errors are assumed to be spatially correlated in the above: $E(\varepsilon_o^i\varepsilon_o^j) = E(\varepsilon_o^i\varepsilon_f^j) = E(\varepsilon_f^i\varepsilon_o^j) = 0$. Equation (9) indicates that the variation of the innovation covariance with $d_{ij}$ is attributable exclusively to the spatial correlation of forecast errors. The forecast error variance $\sigma_f^2$ can be estimated by extrapolating the innovation covariance to the zero separation ($d_{ij} = 0$).

In this study, a least-squares fitting of distance-binned covariance values to a Gaussian function is carried out and the value of

the Gaussian function at the zero separation is assigned to $\sigma_f^2$. Gaussian functions are frequently used to approximate error correlations (Daley, 1991; Gaspari and Cohn, 1999). The algorithm used for the fitting is the bounded and constrained least squares (Lawson et al., 1979). Once $\sigma_f^2$ is determined, Eq. (8) gives $\sigma_o^2$. In essence, HL86 splits the innovation variance into a spatially correlated part ($\sigma_f^2$) and the remainder ($\sigma_o^2$). The error estimates, $\sigma_o$ and $\sigma_f$, over a specific area and period (e.g., within 5° S - 5° N latitude zone and during the months of July) can be diagnosed by applying HL86 to the RO-RO pairs

available within the area and period. Figures 1a-b show the composite distribution of the error estimates: a) bending angle $\sigma_o$ and b) refractivity $\sigma_f$, which are averaged zonally and over the whole data period. The error estimates further stratified into three latitude zones (low, 0-30°; middle, 30-60°; and high, 60-90°) are shown in Figs. 1c-e. The error estimates show a number of remarkable features in the distribution. Not being the focus of this study, however, the features and potential causes are not discussed in this paper. (A separate manuscript is in preparation.) Instead, let it suffice to say that the error estimates show

remarkable spatial variations that must be properly taken into account by regularization methods.

b) Background error correlation

The off-diagonal elements of **B** are diagnosed with the so-called NMC (National Meteorological Center) method (Parrish and Derber, 1992). The method uses the difference between short and long forecasts that are valid at the same time as a proxy for forecast error. Hence, the ECM can be approximated by:

$$\mathbf{B}_{NMC} = E(\delta \mathbf{f} \, \delta \mathbf{f}^{\mathrm{T}}), \tag{10}$$

where $\delta$ is the difference operator between two forecasts of different lead times (12 and 24 hours in this study) and $\mathbf{f}$ is the refractivity sounding modelled with the forecast and placed to a fixed set of RR values. The NMC method does not make use of observations at all. It instead relies on the natural variability of the forecast model. Therefore, the sampling of the forecast difference is not restricted by the availability of RO soundings, meaning that the difference soundings can be taken from every horizontal grid point of the forecast model. The sampling frequency used in this study is 0.5° in latitude and longitude, and 12 hours in time; and, the temporal data coverage is the same 7-year period used for HL86. Again, the ECM over a specific area and period can be estimated by limiting the sampling to the area and period.

While very practical to apply, the NMC method has limitations and is often criticized for lacking theoretical basis. In poorly-observed regions, it underestimates the error variance (Berre, 2000). In addition, the choice of forecast lead times, which affects the size of the forecast difference, is arbitrary at most. Consequently, $\mathbf{B}_{NMC}$ often requires adjustment of the variance (Derber and Bouttier, 1999; Ingleby, 2001). For the reasons, $\mathbf{B}_{NMC}$ is not used in its form in this study. Instead, it is converted to the error correlation matrix $\mathbf{C}$:

$$c_{i,j} = \frac{b_{i,j}}{\sqrt{b_{i,i}} \sqrt{b_{j,j}}} = \frac{b_{i,j}}{\sigma_i \, \sigma_j}, \tag{11}$$

where $b_{i,j}$ and $c_{i,j}$ indicate the elements of $\mathbf{B}_{NMC}$ and $\mathbf{C}$ at $i$-th column and $j$-th row, respectively, and $\sigma_i$ is the square root of $b_{i,i}$, the error standard deviation. Figure 2 shows an example of $\mathbf{C}$ in two latitude bands: a) 5° S - 5° N and c) 70° S - 80° S. These are averaged along the longitude and during the months of July. The profiles of error correlation centred at four arbitrarily chosen heights are shown in b) and d). The error correlation in the tropical latitudes (Fig. 2a) shows oscillatory structures in the stratosphere, which could be related to vertically propagating wave modes that are not well resolved by the forecast model. The exact atmospheric processes behind the oscillation are uncertain for now and an in-depth analysis is underway. Finally, the $\mathbf{B}$ used in this study is modelled as follows:

$$\mathbf{B} = \mathbf{D}^{\frac{1}{2}} \, \mathbf{C} \, \mathbf{D}^{\frac{1}{2}}, \tag{12}$$

where $\mathbf{D}$ is the diagonal matrix of forecast error variance provided by HL86, $\mathbf{D}^{\frac{1}{2}}$ being the square root; and, $\mathbf{C}$ is the correlation matrix diagnosed with the NMC method. For the sake of computational simplicity, $\mathbf{R}$ is assumed to be diagonal.

2.3.3 Implementation

A practical difficulty facing those trying to solve inverse problems of a large size is the computational cost. It is unfeasible to perturb individual elements of the control vector arbitrarily in all directions and sizes, and then search for the very combination of the perturbations that leads to the minimum of the cost function. As mentioned earlier, this study employs the adjoint technique (Lewis and Derber, 1985; Le Dimet and Talagrand, 1986) in order to reduce the cost. The method efficiently computes the steepest gradient of the cost function with respect to all elements of the control vector at once, which is needed for the optimization algorithm used in this study, a quasi-Newton limited-memory algorithm for large-scale optimization (Zhu

et al., 1997). In order to further improve the computational efficiency, the control-variable transform (Parrish and Derber, 1992) is used. To begin with, the incremental form (Courtier et al., 1994) of Eq. (7) is considered:

$$J(\delta\mathbf{x}) = \frac{1}{2}\delta\mathbf{x}^{\mathrm{T}}\mathbf{B}^{-1}\delta\mathbf{x} + \frac{1}{2}(\mathbf{H}\delta\mathbf{x} - \mathbf{d})^{\mathrm{T}}\mathbf{R}^{-1}(\mathbf{H}\delta\mathbf{x} - \mathbf{d}),$$ (13)

where $\delta\mathbf{x} = \mathbf{x} - \mathbf{x}_{\mathrm{b}}$, $\mathbf{d} = \mathbf{y}_o - H(\mathbf{x}_b)$, and $\mathbf{H}$ is the tangent linear version of $H$. The incremental formulation circumvents the nonlinearity of Eq. (7) and reduces computational complexity of the minimization problem. Next, $J(\delta\mathbf{x})$ is reformulated as a function of a new variable, $\mathbf{v} = \mathbf{B}^{\frac{-1}{2}}\delta\mathbf{x}$:

$$J(\mathbf{v}) = \frac{1}{2}\mathbf{v}^{\mathrm{T}}\mathbf{v} + \frac{1}{2}(\mathbf{H}\delta\mathbf{x} - \mathbf{d})^{\mathrm{T}}\mathbf{R}^{-1}(\mathbf{H}\delta\mathbf{x} - \mathbf{d}),$$ (14)

where $\mathbf{B}^{\frac{1}{2}}$ is a square root of $\mathbf{B}$ so that $\mathbf{B} = \mathbf{B}^{\frac{1}{2}}\mathbf{B}^{\frac{\mathrm{T}}{2}}$. The $\mathbf{v}$ representation of the cost function is the actual form used in this study. As a result of the control-variable transform, the background ECM becomes the identity matrix and is thus trivial to deal with (Bannister, 2008). The control-variable transform greatly reduces the conditioning number of background ECM. Consequently, it is easier for the minimization algorithm to find the solution. In practice, VR does not perform $\mathbf{v} = \mathbf{B}^{\frac{-1}{2}}\delta\mathbf{x}$; instead, it carries out the inverse transform $\delta\mathbf{x} = \mathbf{B}^{\frac{1}{2}}\mathbf{v}$, compelling $\mathbf{B}^{\frac{1}{2}}$ instead of $\mathbf{B}^{\frac{-1}{2}}$. This is favourable since it is demanding to invert large matrices. In order to construct $\mathbf{B}^{\frac{1}{2}}$, the method proposed by Kaiser (1972) is used to conduct the eigendecomposition of $\mathbf{C}$:

$$\mathbf{C} = \mathbf{\Sigma}\,\mathbf{\Lambda}\,\mathbf{\Sigma}^{\mathrm{T}} = \mathbf{\Sigma}\,\mathbf{\Lambda}^{\frac{1}{2}}\left(\mathbf{\Sigma}\,\mathbf{\Lambda}^{\frac{1}{2}}\right)^{\mathrm{T}},$$ (15)

where columns of $\mathbf{\Sigma}$ are eigenvectors of $\mathbf{C}$, which are mutually orthogonal ($\mathbf{\Sigma}^{\mathrm{T}}\mathbf{\Sigma} = \mathbf{I}$, where $\mathbf{I}$ is the identity matrix), and $\mathbf{\Lambda}$ is the diagonal matrix of eigenvalues. As $\mathbf{C} = \mathbf{C}^{\frac{1}{2}}\mathbf{C}^{\frac{\mathrm{T}}{2}}$, Eq. (15) gives $\mathbf{C}^{\frac{1}{2}} = \mathbf{\Sigma}\,\mathbf{\Lambda}^{\frac{1}{2}}$. Eventually, $\mathbf{B}^{\frac{1}{2}}$ can be expressed as:

$$\mathbf{B}^{\frac{1}{2}} = \mathbf{C}^{\frac{1}{2}}\mathbf{D}^{\frac{1}{2}} = \mathbf{\Sigma}\,\mathbf{\Lambda}^{\frac{1}{2}}\mathbf{D}^{\frac{1}{2}}.$$ (16)

Since the size of $\mathbf{B}$ used in this study is fairly large ($900 \times 900$), computation of $\mathbf{B}^{\frac{1}{2}}$ at the runtime for each RO event is impractical, particularly for real-time RO data processing. Therefore, $\mathbf{C}^{\frac{1}{2}}$ is precomputed and stored on a $5° \times 5°$ (latitude-longitude) grid for each month of the year, and VR ingests the $\mathbf{C}^{\frac{1}{2}}$ that is nearest to each RO sounding. Moreover, only the largest 100 eigenvalues of $\mathbf{C}^{\frac{1}{2}}$ and the corresponding eigenvectors are retained and stored because the large number of $\mathbf{C}^{\frac{1}{2}}$ files necessitates voluminous storage space. The truncated eigenmodes can replicate $\mathbf{C}^{\frac{1}{2}}$ almost perfectly since the number of modes above the noise floor is generally less than 20. A minor setback is that $\mathbf{C}^{\frac{1}{2}}$ is available on a predefined set of RR values, whereas the lower bound used in VR varies from one RO sounding to another. Therefore, $\mathbf{C}^{\frac{1}{2}}$ and VR differ in the lowest RR.

To deal with the issue, a coordinate of scaled RR is defined as:

$$\eta = \frac{x - x_b}{x_T - x_b},$$ (17)

where $x_b$ ($x_T$) denotes the RR at the bottom (top) of the grid. Next, the background in VR is placed to the $\eta$ grid of $\mathbf{C}^{\frac{1}{2}}$.

The purpose is to reduce the cost by using the $\mathbf{C}^{\frac{1}{2}}$ as it is without any modification. A drawback is that the $\mathbf{C}^{\frac{1}{2}}$ and the background are defined on different grids in terms of RR, which may reduce the accuracy of $\mathbf{C}$ to some degree. The reduced accuracy might be insignificant compared to the uncertainty involved in the diagnosis of $\mathbf{C}$ and is thus considered as a worthy trade-off with the cost reduction. The storage space is no concern for $\mathbf{D}^{\frac{1}{2}}$, which is basically a column vector comprises the diagonal elements. Hence, $\mathbf{D}^{\frac{1}{2}}$ is stored separately in the full resolution. In addition, $\mathbf{D}^{\frac{1}{2}}$ is interpolated to the exact RR values of the background (rather than to the $\eta$ grid) at the runtime.

### 3 Test with a synthetic sounding

It is hypothesized in this study that the major weakness of AI when applied to RO is the uninhibited propagation of measurement error, and the variational regularization is proposed as an alternative. Meanwhile, a common issue that arises while verifying hypotheses with real-world data is that the verifying data accompany their own error, which often impedes drawing decisive conclusions. In order to overcome the difficulty, we begin the verification with a synthetic sounding of bending angle. The synthetic case provides the known true solution against which inversion methods can be verified without any ambiguity. We intend to consider large-amplitude errors so as to emphasize their influence on the solution's quality and to assess the relative robustness of the inversion methods to the erratic observations. This section is also purported as an extended description of the methods but with a tangible example.

### 3.1 Data generation

The tracking of RO signal is often challenging, in particular in the lower tropical troposphere where sharp refractivity gradients frequently exist. As an example, a high-resolution radiosonde sounding, observed at a tropical site in Nauru (0.52° S, 166.93° E) at 12:00 UT on 3 March 2011, shows a complicated structure in the refractivity mainly due to abrupt small-scale moisture variations across multiple inversion layers (Fig. 3). The station was one of the Global Climate Observing System (GCOS) Reference Upper-Air Network (GRUAN) sites until closed in August 2013. The refractivity in the neutral atmosphere can be approximated as (Smith and Weintraub, 1953):

$$N \equiv 10^6(n-1) = k_1 \frac{p}{T} + k_2 \frac{p_w}{T^2}, \tag{18}$$

where $T$ is temperature in K; $p$ is (total) pressure in hPa; $p_w$ is water vapor pressure in hPa; and, $k_1$=77.6 hPa K$^{-1}$ and $k_2$=3.73·10$^5$ K$^2$ hPa$^{-1}$ are coefficients.

The discretization error of AI is significant when the resolution of the measured bending angle is poor. Therefore, the radiosonde sounding in the highest resolution available to us is used for the simulation of the bending angle. The intent is to improve the discrete AI, because the discretization error of VR relates to the resolution of the background. The sampling rate of the radiosonde data is one second in time, which corresponds to about 5 m on average in height interval. The bending angle simulated with the radiosonde data and using Eq. (4) also presents rapid variations (green line in Fig. 4a). This bending angle

is assumed to be of absolute accuracy (error-free) and is referred to as the perfect (true) observation. The measurement error of the radiosonde data carried forward into the bending angle is considered as legitimate small-scale variations in the true atmosphere. After that, a synthetic observation is generated by adding a suppositional measurement error to the perfect observation. The measurement error $\varepsilon$ is assumed to follow a first-order autoregressive process and modelled as:

$$\varepsilon_k = \sigma_k \mu_k = \sigma_k \left( \rho_{k,k-1} \mu_{k-1} + \eta_k \right), \tag{19}$$

where $\sigma$ is the statistical measurement error (Fig. 1a) at the time and location of $\varepsilon$; $\mu$ and $\eta$ are random normal (zero-mean and unit-variance) variables; subscripts indicate the height indices of measurement samples (top-down order with increasing $k$); and, $\rho_{k,k-1}$ is the error correlation between the two height levels. The error correlation is modelled with the Gaussian function:

$$\rho_{k,k-1} = e^{-\frac{(a_k - a_{k-1})^2}{2L^2}}, \tag{20}$$

where $a$ is again the IP and $L$ is the length scale of error correlation. Assuming that the measurement error correlation is very weak, we set $L$ to 10 m. The black line in Fig. 4a is the error-added, "measured" bending angle.

### 3.2 Abel inversion of low-pass filtered bending angle

Noise in the measured bending angle negatively affects the quality of RO refractivity, unless properly mitigated. It is thus customary to smooth the bending angle prior to AI. In order to mimic the practical application of AI, a low-pass filtering, the

fourth-order Butterworth filter (Butterworth, 1930) with a cutoff wavelength of 200 m, is applied to the measured bending angle (red line in Fig. 4b). The following step is obtaining refractivity soundings through AI from the true, measured, and filtered bending angles. It must be mentioned that the resulting true refractivity (i.e., the one derived from the true bending angle via AI) differs slightly from the refractivity used to generate the true bending angle (shown in Fig. 3c). The discrepancy stems from the numerical approximations made for the analytical AI.

Figure 4c shows the refractivity errors, which are the differences from the true refractivity. The red line indicates the refractivity error when the filtered bending angle is used for AI. The result with the "raw" bending angle (black line) is shown as the reference against which the effect of the low-pass filtering can be evaluated. A common problem with any low-pass filtering is that the degree of smoothing is hard to control. An excessive smoothing leads to a loss of observational information, whereas a minor filtering causes insufficient noise attenuation. On top of that, measurement noise is often non-stationary,

meaning that the noise spectra vary with height in accordance with vertically varying atmospheric structure. For instance, the low-pass filtering used in this study tends to reduce the refractivity error below 2 km and above 6 km, where the true bending angle varies rather slowly. On the contrary, the filtering increases the refractivity error around the local peaks of the bending angle in 2-6 km height range. As different occultation events encounter different atmospheric conditions, it is impractical to design a customized low-pass filter for each occultation that is adaptive to the local noise spectrum that varies with the height.

Another limitation of the sequential approach (i.e., filtering followed by AI) is the one-way flow of information in the process. That is, AI does not pass any information about the effect of unattenuated noise back to the filtering. The penalty term of

regularization methods acts like a low-pass filtering. The difference is that the penalty term invokes a reverse communication about the perceived noise power while the term is minimized iteratively and jointly with the data fidelity term. In VR, the feedback to the control vector is given through the adjoint of the observation operator.

### 3.3 The errors-in-variables problem

The EIV problem occurs when the independent variable (position in space or time) of measurements is not known perfectly. Suppose that a particular type of radiosonde system has an offset error in the height. In that case, a flawless temperature sensor of the system is bound to produce a temperature bias, which is the height offset multiplied by the local temperature lapse rate, when monitored at a fixed height. The EIV error also emerges while comparing a measurement with its model, if the independent variable of the measurement is not one of the model's coordinates. This is exactly the case for RO bending angle.

The location of RO bending angle is defined with the IP and the model counterpart is the RR. As explained in Sect. 2.3, the modelled RR cannot be perfect because the model refractivity is never perfect. The RR error increases the discrepancy between measured and modelled bending angles, which is interpreted as a measurement error from the model's perspective. This issue is closely relevant to the data assimilation of RO bending angle, at least for the methods in which the control variables are defined in the model space. The physical-space statistical analysis (Cohn et al., 1998) can be an exception but is not considered

in this study because model-space methods are currently in prevailing use for operational weather forecasting.

The EIV error in bending angle $\varepsilon_\alpha$ can be estimated as: $\varepsilon_\alpha = \frac{d\alpha}{da} \cdot \varepsilon_a$ ; $\varepsilon_a = r\varepsilon_n$, where $\varepsilon_a$ and $\varepsilon_n$ are model's error in the IP and refractive index, respectively. In this study, the error estimate of 12-h ECMWF forecasts in the refractivity ($\sigma_N^b$) (Fig. 1b) at the time and location of the synthetic sounding is considered as the representative of $\varepsilon_n$ (Fig. 5a). The corresponding $\varepsilon_a$ is shown in Fig. 5b and the black line in Fig. 5c indicates $\varepsilon_\alpha$, where the bending angle gradient is based on the true observation.

The dashed red line in Fig. 5c indicates the statistical observation error in the bending angle ($\sigma_\alpha^o$) shown in Fig. 1a. Overall, the EIV error ($\varepsilon_\alpha$) is comparable to the statistical estimate ($\sigma_\alpha^o$) in magnitude for the particular synthetic sounding used in this study. However, $\varepsilon_\alpha$ is exceedingly larger than $\sigma_\alpha^o$ at the heights of sharp bending angle gradient. As explained in the introduction, the EIV error makes the true bending angle inaccessible to RO data assimilation. What is visible to the data assimilation is the one to which $\sigma_\alpha^o$ is added. As a result, the assimilation of the true bending angle cannot yield the true

refractivity. On the other hand, the VR with the true observation is able to reproduce the perfect refractivity at least hypothetically, because the method holds the solution and observation in the same space. Namely, the perfected bending angle in the IP space is the sufficient condition for the perfect reconstruction of the refractivity.

### 3.4 The proposed variational regularization

Figure 6a shows the trace of the cost function of VR with iteration. Because the initial solution is the same with the background,

the background term $J_b$ (dashed blue line) starts from zero and gradually increases with iteration as the solution deviates more and more from the background. The observation term $J_o$ (thick red line) decreases rapidly with iteration as the mapped solution

approaches the observation. The iteration continues as long as the total cost function $J_T$ (thick black line) keeps decreasing and a norm of $\nabla_{\mathbf{x}} J_T$ falls below a prescribed threshold. In the case considered here, all of the cost functions are nearly flat after 15 iterations. Figure 6b compares errors in the refractivity. The background error $\sigma_N^b$ (dashed blue line) is sizable at a number of heights. That is mainly because the vertical resolution of the 12h ECMWF forecast used here (91 levels) is insufficient to

represent all small-scale details of the true refractivity, especially the local peaks (Fig. 3c). The proposed method (red line) yields a refractivity error considerably smaller than $\sigma_N^b$. The heavy black line is the refractivity error resulting from the AI with the "measured" (rather than the smoothed) bending angle, which is the same as in Fig. 3c and overlaid for comparison. VR is smaller than the AI in the refractivity error almost everywhere. As described earlier, the low-pass filtering shown in Fig. 4b is unable to reduce the refractivity error (Fig. 4c) and the use of different cutoff wavelengths does not make any notable difference

in the result. The pre-filtering is ineffective in the error reduction at least for the particular synthetic sounding used in this study, where the true bending angle contains high frequencies that cannot be isolated from the noise. Although VR is able to cut down the refractivity error by more than half compared to AI, it is difficult to further reduce the remaining error because the measured bending angle is severely corrupted here and there by the large-amplitude noise. It is worth noting in Fig. 6b that VR approaches AI, deviating significantly from the ECMWF forecast. This suggests that the influence of the background on

the solution of VR is minor, as long as the observation is unbiased and of good quality in the larger-scale perspective.

### 3.5 Posterior height determination

In our experience, RO refractivity tends to agree with correlative data better than the bending angle (used to derive the refractivity) suggests. For instance, the comparison of RO data with independent verifying data (e.g., short-term NWP forecast or high-resolution radiosonde observation) can be made separately in bending angle and refractivity. The comparison provides

$\Delta\alpha = \alpha^o - \alpha^m$, where $\alpha^o$ and $\alpha^m$ indicate the observed and modelled bending angle, respectively. Likewise, the comparison in the refractivity gives $\Delta N = N^o - N^m$. Once available, $\Delta\alpha$ and $\Delta N$ can be compared to each other. For example, $\Delta\alpha$ can be propagated into the refractivity by: $\Delta N_\alpha = \langle \mathbf{H} \rangle^{\mathrm{T}} \Delta\alpha$, where $\langle \mathbf{H} \rangle^{\mathrm{T}}$ denotes the linearized AI that includes the conversion between $n$ and $N$. As said in the beginning, $\Delta N$ is generally smaller than $\Delta N_\alpha$ in magnitude. The same is observed in the comparison of error estimates (e.g., Fig. 1a-b). For instance, the bending angle error propagated into the refractivity by means of the Monte

Carlo approach is larger than the refractivity error that is estimated separately. The reason that $\Delta N$ is smaller than $\Delta N_\alpha$ relates to the way that the height of derived refractivity is determined in AI. As can be understood from Eq. (5), AI provides the refractive index as a function of the IP, $n = n(a)$. Afterward, the height is determined by: $z = an^{-1} - R_c$. In the above-mentioned examples that convert $\Delta\alpha$ into $\Delta N_\alpha$, on the other hand, the height of the refractivity is predefined and does not change afterwards. That is because the conversion is based on linear approximations and so the perturbations do not change

the location assigned to the variables as well as the reference state. In that regard, data assimilation methods are the same because the location of the solution (state variables) is kept unchanged during the minimization of the cost function. Otherwise, the cost function fails to remain consistent in the course of the minimization. As described in Sect. 2.3.1, VR also uses the

Posterior Height Determination (PHD). The control variable in VR is defined as a function of RR. After the solution is found at the minimum of the cost function, VR determines the height using the optimal refractive index and the RR.

The PHD may sound trivial, but it has a substantive effect on the interpreted quality of the solution. An example is illustrated in Fig. 7. The heavy solid line in Fig. 7a is the sounding of true refractivity. The solution will appear somewhere on a horizontal line at the true height of a given IP, deviating arbitrarily from the true refractivity. (In this context, RR and IP are the same in the meaning; hereafter, IP is thus used preferably for the sake of simplicity.) The example considered here is the horizontal line with open-headed arrows around 2.8 km and next to $N(a)$. The solution's error is indicated by the distance from the true refractivity (i.e., the open circle in the middle of the line). Now suppose that the solution is smaller than the true refractivity at the given IP. Because $r = an^{-1}$, PHD places the solution at a higher location in the height coordinate than the true height. Likewise, a positive refractivity error pushes down the solution. As a result, the trajectory of possible solutions is slanted as shown by the dashed line next to $N(r|a)$.

An example inside the small box is shown in Fig. 7b to offer a detailed depiction, where the solution (denoted **P**) is smaller than the true refractivity (denoted **T**) by an error $\delta N < 0$. It can be shown that the height displacement of the solution due to PHD is: $\delta r \simeq -r\delta n = -10^{-6}r\delta N$. The solution is thus placed at a higher location ($r'$) than the true radius ($r$). What is important here is that the solution's error in the height coordinate is to be perceived as the difference from $N(r')$ rather than from $N(a)$, because the solution's true height is never known in the real world. For the particular example shown in Fig. 7b, the solution's apparent error $\overline{|\mathbf{P'T'}|}$ is smaller than that the true error $\overline{|\mathbf{PT}|}=\delta N$, where the overbar denotes the line connecting two points and vertical bars indicate the line length. By linearizing $a = nr$, the slope of $\overline{|\mathbf{P'T}|}$ can be shown as:

$$\frac{dN(r|a)}{dr} = -\frac{10^6 n}{r} \simeq -157 \text{ km}^{-1}, \tag{21}$$

where $r|a$ stands for the conditional radius of the solution given the IP. The slope is indeed the critical refractivity gradient at which the ray's curvature radius is equal to the Earth's radius and is thus the threshold for the occurrence of the critical refraction. By inspection, the solution's true error $\delta N = G_c \delta r$ and the apparent error $\delta N' = \overline{|\mathbf{P'T'}|} = (G_c - G)\delta r$, where $G_c$ is the critical gradient ($-157 \text{ km}^{-1}$) and $G$ is the local refractivity gradient. Given that $\frac{\delta N'}{\delta N} = 1 - \frac{G}{G_c}$, $\delta N'$ is smaller than $\delta N$ in the absolute size if $2G_c < G < 0$. In general, PHD reduces the apparent error since $G$ is known to be about -40 km$^{-1}$ in the standard atmosphere (United States, 1946). An example for which the apparent error is larger than the true error is shown at the point denoted as **B** in Fig. 7a, where the true refractivity increases with height. It must be underscored here that not only the derived refractivity but also the height assigned to it possesses error; and, the errors are negatively correlated. Therefore, the derived refractivity is also subject to the EIV problem. However, in practical circumstances (e.g., when a comparison to other data is made), the errors are attributed entirely to the refractivity and the height is assumed to be free of error.

Figure 8 illustrates the response of $\frac{\delta N'}{\delta N}$ to varying $G$. The solid black line in Fig. 8a is the true refractivity sounding in the IP coordinate and black dashed line indicates a hypothetical solution in the same coordinate, which is set to be 5% larger than the true solution everywhere. In the example, the gradient of the true refractivity, $G$, ranges from -135 km$^{-1}$ to 22 km$^{-1}$ (Fig. 8b).

The red lines in Fig. 8a are the same refractivity soundings but seen in the height coordinate. Again, the solid line and dashed line indicate the true refractivity and the solution, respectively. As shown in Fig. 8c, $\delta N'$ is less than 5 % ($\delta N$) except for around 3.25 km where $G$ is positive. In particular, $\delta N'$ is significantly smaller than $\delta N$ near 2.3 km where $\delta N' \approx 0$ as $G \approx G_c$. The tracking of RO signals that are affected by the critical refraction, which occurs when $G$ is negatively large, is known to be

challenging. In the heights that $G \leq G_c$, therefore, the quality of RO refractivity is not expected to be the best. Surprisingly, however, Fig. 8c shows an opposing result: PHD results in the smallest refractivity error when $G \approx G_c$. That is because PHD purges all the apparent error no matter how big the true error is. On the other hand, PHD increases the apparent error in case of strong sub-refraction ($G > 0$), which is often observed at the immediate underside of local refractivity peaks. Therefore, the apparent refractivity error depends on the optical structure in the atmosphere as well as quality of the bending angle data.

Syndergaard (1999) described the reduction of refractivity error resulting from PHD but without relating it to $G_c$.

## 4 Test with real data

In this section, we apply AI and VR to actual RO events and compare the resulting refractivity soundings with nearby radiosonde observations. In doing so, we use two sets of radiosonde data in order to complement each other's weakness. In the following, the data sets used here (including RO, NWP, and radiosonde) are briefly described and next the validation with

respect to the radiosonde data is presented.

### 4.1 Data

The GPS RO data used in this study are made available from the Constellation Observing System for Meteorology, Ionosphere, and Climate (COSMIC) mission and are processed by the CDAAC. The six COSMIC satellites have been producing 1,000-2,500 globally distributed soundings each day since the launch in April 2006 (Anthes et al., 2008). Kuo et al. (2004) and

Schreiner et al. (2011) describe the CDAAC's data processing. VR takes the COSMIC neutral atmospheric bending angle as the input, as does CDAAC's AI. This is to ensure the consistency between the methods in the observation. In addition, we take CDAAC's refractivity as the solution of AI, instead of carrying out an AI ourselves, in order to avoid the potential uncertainty involved in the practical implementation of AI. Hence, the CDAAC's refractivity is considered as the standard solution obtainable from the CDAAC's bending angle and via AI. The COSMIC data used here (version 2013.3520) are available

online at http://cdaac-www.cosmic.ucar.edu/cdaac/products.html. The background soundings of VR are generated from the operational ECMWF forecasts, which are on a reduced Gaussian grid (~25 km spacing in latitude and longitude) with 91 vertical levels between the earth's surface and 80 km. The same resolution is used throughout the study period for the sake of convenience, although the model's spatial resolution has been increasing with time (e.g., ~9 km and 137 levels as of March 2017). When the upper bound of Abel transform is not high enough, the integral becomes negatively biased in high altitudes.

To reduce the bias, the empirical model of the US Naval Research Laboratory (NRL), MSIS (Hedin, 1991) are used to provide the refractivity above the ECMWF model top up to 2,000 km.

One of the radiosonde data sets used in this study is the Automatic Data Processing (ADP) upper air observation provided by the Data Support Section (DSS) of UCAR (available online at http://rda.ucar.edu/datasets/ds337.0). The data are the global six-hourly upper air reports routinely collected by the National Centers for Environmental Prediction (NCEP) for operational uses through the Global Telecommunications System (GTS). The reports consist of messages that are prepared using a set

of World Meteorological Organization (WMO) alphanumerical TEMP (upper air soundings) codes, e.g., FM-35 (land stations) and FM-36 (ship-based). The codes were designed to keep the messages as short as possible whilst retaining all noteworthy features observed during the balloon's ascent. As a result, TEMP codes support a limited vertical resolution, allowing reports only on the standard (mandatory) pressure levels and significant levels (if there is any). The other set of radiosonde observation is the high vertical resolution data from the radiosonde stations operated by the National Oceanic Atmospheric Administration

(NOAA), available online at ftp://ftp.ncdc.noaa.gov/pub/data/ua/rrs-data/. Beginning in 2005, NOAA began transitioning from radiotheodolite balloon tracking to GPS tracking. The data from this new system, called Radiosonde Replacement System (RRS), are recorded at 1-second resolution, permitting good representation of small-scale atmospheric structures. A particular advantage of this data set regarding the comparison to RO data is that it provides balloon's height at every recorded moment of the flight. The operational radiosonde data (ORD hereafter) on the other hand have height reports only on the standard

pressure levels. Thus, the heights on significant levels must be estimated based on the measured values of pressure (p), temperature (T), and humidity (U). The reconstructed heights are of suitable quality in general, but at times have larger uncertainty due to the poor data resolution as well as measurement error in pTU. The pTU-height error is interpreted as a refractivity error when the radiosonde data are compared to RO refractivity in the height coordinate, which is another EIV problem. Using the high vertical resolution radiosonde data (HVRRD), the pTU-height error on significant pressure levels can

be avoided.

## 4.2 Validation with operational radiosonde data

ORD has a larger number of soundings and a superior geographical coverage compared to HVRRD, the stations of which belong to a small subset of ORD sites. Hence, ORD provides more soundings that are closely located (collocated hereafter) with RO soundings for a given period of time. It also allows the collocated soundings to be sampled at various locations and

under diverse atmospheric conditions. Focusing on tropical and subtropical regions, we used 24,328 collocated (< 2 h and < 200 km) ORD-COSMIC matches in latitudes between 35°S and 35°N for the period from 17 February 2007 to 7 November 2010 (Fig. 9). The radiosonde data that are unphysical or deviate unrealistically far from ECMWF forecasts are discarded prior to the validation through a series of data screening as done by Wee and Kuo (2014). Also, COSMIC soundings flagged as bad by CDAAC are dropped.

The particular latitude zone is chosen to ease the comparison between AI and VR. However, the sharp refractivity gradient in the lower troposphere over the regions could cause the critical refraction, which results in a significant negative bias in the AI-produced refractivity all the way down to the surface from the top of the ducting layer. The critical refraction thus gives a serious penalty to AI. While the AI-produced refractivity in the heights affected by the critical refraction tends to appear as

outliers in the comparison, the PHD described in Sect. 3.5 makes the outlier detection futile. Therefore, the critical refraction makes the statistical comparison more difficult. For this reason, this study attempts to detect the occurrence of critical refractions for each RO sounding and exclude the affected heights, if they exist, from the comparison to ORD. This step is considered as a quality control. To do so, ducting layers are searched in the background ($G < -150$ km$^{-1}$, slightly relaxed from $G_c$) starting from 7 km and downward. The procedure is repeated with ORD sounding, but not with AI sounding because the refractivity gradient of AI cannot be smaller than $G_c$. When a ducting layer is detected for the first time, the refractivity below the layer's top is discarded from the comparison. In VR, the top of the ducting layer becomes the lower bound of the computational domain. Consequently, both the background and measured bending angle below the top of the ducting layer are not used in VR. Needless to say, the refractivity comparison to ORD is limited to the height range common to AI, the background, and VR. In order to make the comparison robust to outliers, the lower and upper 1 % of AI soundings in the difference from ORD are discarded at each height level. The matching soundings of VR and the background are also rejected. Figure 10a compares the difference from ORD in the mean refractivity. AI (thick dark grey line) results in a distinct negative bias below 3 km, which increases with decreasing height reaching -1.5 % near the surface. This means that the above-mentioned quality control is not perfect although it has reduced the maximum bias from -3.5 % (not shown). The persisting bias might be due to sub-critical refractions or some ducting layers undetected by the forecasts and ORD. AI shows a small positive bias above 4.5 km, which are about 0.2 % at 6.5 km. The refractivity bias of AI mainly stems from the bias in the observation. What intensifies the negative refractivity bias in the lower troposphere is the downward propagation of the observation bias. On the other hand, 12h (solid blue) and 24h (dashed green) ECMWF forecasts in that height range deviate very little from ORD. In the lowest 2 km, the forecasts show positive systematic deviations that increase with the forecast lead time. Considering that moisture is the dominant contributor to the variability of refractivity in the height range, the ECMWF forecast model is likely to have a wet bias in the planetary boundary layer as shown by Flentje et al. (2007), at least with respect to ORD. In both cases that 12h (solid red) and 24h (dashed gold) forecasts are used as the background, VR is less biased than AI throughout the entire height range. In particular, the negative refractivity bias below 4 km is greatly reduced. This is reasonable because VR does not propagate the negative measurement bias downward. Moreover, VR can reduce the effect of measurement bias with the aid of background, especially for the exceedingly large biases around the local peaks of the bending angle and when the background is largely unbiased, like the ECMWF forecasts used here. Overall, VR is in between AI and the background and is closer to the background in the lowest 4 km. VR approaches the background in the mean because doing so describes the measurement better. As shown by the synthetic data test, the influence of the background on VR is very limited when unbiased measurement is given (Fig. 6b). With biased measurement, therefore, the approach of VR to the background is desirable.

In the standard deviation from ORD (Fig. 10b), AI is slightly larger than 24h forecast in the lowest 2 km and is comparable elsewhere. Thus, AI is no better than 24h forecast. As expected, the random error of the forecast increases with the lead time. Encouragingly, VR is smaller than the background throughout the height range regardless of the forecast lead time. This indicates that VR works as designed attaining an error variance smaller than those of observation (represented by AI) and the

background. Nevertheless, it is noticeable that AI, FCST, and VR are similar in magnitude and structure, and the difference among them is very small, less than 0.5 % at most. The main reason behind the similarity is the large error of ORD, which is shared by AI, FCST, and VR. An error estimation in which HL86 is applied to nearby COSMIC-ORD pairs for the same 7-year period described in Sect. 2.3.2 suggests that the refractivity error variance of ORD is slightly larger than that of AI (not shown). It means that the actual errors of AI, FCST, and VR are about 70 % of the standard deviations shown in Fig. 10b. The differences among the three appear small because those are added on top of the same large ORD error. Another reason for the large standard deviations (and small differences among them) is the spatiotemporal distance between COSMIC and ORD soundings. Although the collocation threshold used in this study is reasonably tight, the significant horizontal inhomogeneity (especially in moisture) over the tropical region causes the two nearby soundings to differ considerably from each other. The difference is particularly large if the small-scale features in those two soundings are out of phase. The systematic difference on the other hand is insensitive to the distance.

The vertical resolution of verifying observation is also relevant to this issue. As mentioned earlier, ORD has a low resolution and the data points are distributed irregularly in height. Moreover, the depths between adjacent data levels differ substantially from one sounding to another depending on the number and location of significant pressure levels that are reported. The significant levels are where the observed atmospheric structure turns or changes abruptly. The rapid change around the significant pressure levels, in conjunction with the limited data resolution, results in sizable error if one attempts to interpolate ORD to a regular height grid. Therefore, all other correlative data (AI, FCST, and VR) are interpolated to the heights of ORD in this study. Afterwards, the data points are binned according to the pressure of each ORD height for the statistical comparison. For standard pressure levels, all data samples of the same pressure are grouped together and assigned to the pressure level. For significant levels on the other hand, bins are allocated in the middle of adjacent standard pressure levels, accommodating all data samples whose pressure are between the neighbouring standard levels. For instance, all significant pressure levels between 700 hPa and 850 hPa are assigned to the same bin. For this reason, the data counts exhibit a saw-toothed distribution as shown in Fig. 10c. For each bin, the mean refractivity and mean height are represented by those of ORD. The shortcoming of the binning approach is that the samples in a bin are different in height. Given that the refractivity varies exponentially with height to a good degree, the height discrepancy increases the samples' spread (standard deviation), which could have been reduced greatly if all the samples were taken at the same height. Namely, a good portion of the standard deviation shown in Fig. 10b is attributable to the vertical variation of the true refractivity. In other words, some of the standard deviation is the EIV error, which is caused by disregarding the height difference among individual samples. Another issue with the binning is that the statistics is sensitive to the way that the binning is done. For example, the statistics obtained using a height-based binning (not shown) appeared different from those shown in Fig. 10. That was more evident in the mean difference, which is greatly influenced by the distribution of the heights of individual samples in each bin. While the binning approach is compelled by the poor vertical resolution of ORD, it is not so dependable introducing substantial extra uncertainty (i.e., EIV error) to the statistics. A correction accounting for the inter-sample height difference might be possible, but no attempt in the direction with ORD data is made in this study. These issues with the binning can be addressed by using radiosonde data of higher resolution.

**4.3 Validation with high vertical resolution radisosonde data**

In this comparison to HVRRD, all data sets (HVRRD, AI, FCST, and VR) are interpolated to a regular height grid of 50-meter interval. Thanks to the high vertical resolution, the interpolation of HVRRD does not cause large error unless the reported heights are corrupted. Besides, data binning is unnecessary since all data samples are placed in the same height. Therefore, the

uncertainty arising from the inter-sample height discrepancy (i.e., EIV error) is eliminated. Figure 11 shows the location of 92 HVRRD stations used in this study. The stations are classified into three regions: the tropics (15 sites, red filled circles), US (64, green open diamonds), and Arctic (Alaska) (13, blue filled squares). With the same distance threshold used for ORD, 30,796 collocated COSMIC-HVRRD pairs (2,602 in the tropics; 25,128 US; and, 3,064 Arctic) are found for the period from 17 February 2007 to 31 August 2015 (insets in Figs. 12a-c). For COSMIC data, two slightly different data versions are used:

2013.3520 through 30 April 2014 and 2014.2860 afterwards. Without the smearing due to the binning, the comparison shows a more detailed vertical structure of the statistics. Another difference from the comparison to ORD is that the detection of the critical refraction is carried out with only the background. The reason for not using HVRRD-based detection is that the derivative between very shallow layers (due to the high resolution) intensifies measurement noise and leads to rapidly oscillating refractivity gradients.

Figures 12a-c compares the refractivity biases with respect to HVRRD in the three regions. In the tropics (Fig. 12a), AI again shows a large negative bias near the surface and a moderate positive bias in 6-10 km range. Without the spread due to the data binning and the HVRRD-based detection of critical refractions, the peak values of the bias (-2.6 % near the surface and 0.5 % at 7 km) are larger in magnitude than those obtained by the comparison to ORD. The background, 24h ECMWF forecast (denoted as FCST), is biased negatively below 6 km and positively in 5-10 km range. This is quite different from the

comparison to ORD, where the background shows a positive bias near the surface. This may indicate the uncertainty of these radiosonde data in the mean refractivity. The two radiosonde data sets used in this study, ORD and to HVRRD, differ in a number of aspects. One is the geological coverage of the observation network. The HVRRD stations are smaller in the number and located in small specific areas. For instance, about 10 out of 15 HVRRD stations are in latitudes lower than 20°, whereas the majority of ORD sites are located in the subtropics. Therefore, the atmospheric conditions over the two sets of radiosonde

station differ, leading to different error characteristics of RO bending angle. The same is true for the data quality of AI-produced refractivity and the background. The radiosonde data sets also differ in the instrumentation and reporting practice. For instance, the height reports in HVRRD are based on GPS tracking, whereas those in ORD are a mixture of the pTU height and the GPS-based height. Needless to say, the vertical resolutions of ORD and HVRRD are different. The comparison to ORD is subject to extra uncertainty due to the data binning and the subsequent EIV error. All these factors and others contribute

to the difference in the assessed statistics. Nonetheless, the high vertical resolution is an irrefutable strength of HVRRD in regard to the verification of RO data. Hence, the comparison to HVRRD is believed to provide trustworthier results in this study. That being said, it is remarkable that VR follows AI very closely above 2 km, deserting the background. In the lowest 2 km where AI is greatly biased, VR shows a diminished negative bias. This again indicates that the influence of the

background on VR is not worrisome unless the observation is biased. The closeness between AI and VR also suggests that the common deviation from HVRRD is indicative of the systematic error of HVRRD. The bias of AI in the lower altitudes is near neutral over the US (Fig. 12b) and turns into positive over Alaska (Fig. 12c). On the other hand, the bias in the middle troposphere remains positive and does not change much in the magnitude. The differences among AI, FCST, and VR are small over the US and the three are almost on top of each other over Alaska. Despite the smaller differences, VR is less biased than AI in all heights and in both regions. In higher altitudes, AI deviates a bit more from VR. As shown by Wee and Kuo (2015), the bias of AI in the stratospheric heights might be caused by inter-annual atmospheric variations that are unrepresented by the climatology used for the statistical optimization of the bending angle.

The difference among AI, FCST, and VR is more pronounced in the standard deviation (Figs. 12d-f). In the comparison to ORD (Fig. 10b), AI is no better than the 24h ECMWF forecast especially in the lowest 2 km. In the standard deviation from HVRRD, on the other hand, AI is smaller than the forecast in the tropospheric heights of all regions. This suggests that the EIV error caused by the data binning is large enough to bring forth a misleading conclusion in the comparison to ORD. In the stratosphere above 13 km over the US and Alaska, AI is slightly larger than the forecast. It is found through the error estimation described in Sect. 2.3.2 that the stratospheric degradation of AI is due to a rapid, unsmooth transition of bending angle estimation, which is from the geometrical optics method applied above 20 km to a wave optics method below. Nonetheless, VR agrees better with HVRRD than the others do in all heights regardless of the geographical area. As a result of the high vertical resolution, HVRRD has small-scale vertical variations in the refractivity, which might not be easy to be observed concurrently by other observing systems. Therefore, the evident error reduction attained by VR in both measures (i.e., systematic and random errors) is very impressive.

**5 Summary and concluding remarks**

The refractivity soundings provided by GPS RO have been used widely for weather and climate research. Typically, the refractivity is obtained from the inverse Abel transform (Abel inversion) of measured bending angle (measurement). The foremost problem of Abel inversion (AI) is that it allows the measurement error to propagate downward freely. The measurement error includes artefacts introduced by arbitrary noise mitigations that are applied prior to AI. It is challenging to improve the noise mitigation because the separation of signal components and noise is never easy. After considerable deliberation, we come up with an idea that it is synergetic to combine noise attenuation and refractivity inversion together into an estimation problem. Another contemplation is that the issue of measurement error propagation can be addressed by instead using the forward Abel transform (FAT). As the realization of these hypotheses, a variational regularization (VR) of the FAT is proposed in this study.

The proposed method considers the numerical inversion of the FAT. Doing so does not require the vertical integration of error-possessing measurement and precludes the measurement error propagation that is the root cause for the degradation of AI-produced refractivity. While AI considers the measurement to be complete, VR regards it approximately accurate. Hence, instead of solving the inverse problem directly, VR turns it into an optimization problem in which the fitting to the

measurement is used as a weak constraint, while the solution's behaviour is regularized as per the prior information. The optimization problem is solved via the adjoint technique, which is a very efficient way of calculating the gradients of the cost function with respect to all control parameters at once. The essential feature of VR in the formulation that differentiates the method from classical regularizations is the use of error covariance matrices (ECMs), which permits a rigorous incorporation

of prior information on measurement error characteristics and solution's desired behaviour into the regularization. The proposed method needs a first guess to kick off. This study considers NWP forecasts to be the most adequate as the first guess because they are of good quality and routinely available, and offer reliable error estimates of the observation as well as themselves that in turn support the construction of realistic ECMs. The specific first guess used in this study is short-term operational forecasts of the ECMWF. The diagonal elements of the ECMs are estimated by applying the Hollingsworth-

Lönnberg method to closely located pairs of RO soundings, whereas off-diagonal elements of the forecast ECM are approximated by employing the NMC method. The observation ECM is on the other hand assumed to be diagonal for the sake of computational simplicity.

The regularity imposed on the solution is accomplished through the filtering effect of the background ECM, which is controlled by the off-diagonal elements (spatial error correlations) that spread information from each measurement sample to

neighbouring locations. In addition to being smooth, the solution of VR attains the statistical optimality delineated by the ECM. This study limits the scope of the proposed method to the relationship between bending angle and refractivity, in order to circumvent additional uncertainty and complication that give to rise when the regularization problem is extended to other variables (i.e., temperature, pressure, and moisture) in such methods as RO data assimilation and 1D-Var. A unique feature of VR in that respect is the coincidence of the solution space with the data space that eliminates the ambiguity resulting from the

coordinate transform between the refractional radius (RR) and the height, which is inevitable in the RO data assimilation and 1D-Var. The significance of having the solution and observation in the same RR space is that it permits the perfect retrieval of refractivity from error-free measurements, which is unviable for RO data assimilation and 1D-Var. That is, the flawless replication of measurement in the RR coordinate is the sufficient condition for the perfect refractivity retrieval. In that sense, VR can be understood in that it enhances the measurement, aided by the background, with the regularized refractivity as the

consequence.

The proposed method along with AI is tested by means of a synthetic sounding with error. The known true solution in the controlled setting resolves a long-standing problem in real-data tests, which is the ambiguity stemming from the uncertainty of verifying data. The weakness of AI is demonstrated with the focus on the effect of measurement error propagation.

It is shown with an example that the smoothing of measured bending angle prior to AI does not reduce the refractivity error

when the measurement is corrupted with large-amplitude, non-stationary noise. The errors-in-variables (EIV) problem is identified as a notable source of measurement error for RO data assimilation and 1D-Var. At the heights that the bending angle varies rapidly, the EIV error is revealed to be exceedingly larger than the statistical measurement error. Another point highlighted in the test is the posterior height determination (PHD). It is described in detail with examples and illustrations that PHD reduces the refractivity error substantially. We have utilized the synthetic case in order to articulate the reason that we

had decided to define the control variable of VR in the observation space. That is to reduce the EIV error and to take advantage of PHD. The test with the synthetic data has demonstrated that VR is able to yield an accurate solution that is superior to the AI-produced refractivity. An important finding is that the solution of VR approaches the true solution deviating from the background of sizable error, once the observation provided is unbiased.

The proposed method is applied to actual COSMIC events and the result is validated with nearby radiosonde soundings. For the validation two radiosonde data sets, the operationally collected TEMP-format global data and the high vertical resolution data collected at stations operated by NOAA, are used to complement each other's weaknesses. The former is lower in the resolution but has a superior geographical coverage, and vice versa. The validation shows that the standard deviation of refractivity from the radiosonde data is persistently smaller with VR than with AI and the background. Both radiosonde data

sets equally show the smaller standard deviation of VR in all heights and latitudes. VR also agrees better with the radiosonde data than AI does in the mean of refractivity, especially in the lowest 2 km. We have seen in some heights that VR is slightly larger than the background in the mean difference. Although not certain for sure, the likely cause is the systematic error of the radiosonde data. It is found that the limited vertical resolution of the TEMP-format radiosonde significantly reduces the adequacy of the data set for a precise verification of RO data. The comparison to the high vertical resolution radiosonde data

confirms that the influence of the background on the VR-retrieved refractivity is minor in the heights and regions that the systematic error of RO bending angle is relatively small. Even in the lowest few kilometres that AI-produced refractivity has large negative bias, VR reduces the refractivity bias considerably by preventing measurement bias from propagating downward.

Based on the results presented herein it is concluded that VR is a considerable improvement over AI in the quality of

refractivity. This suggests that VR is able to enhance the data value of RO bending angle with the aid of prior information. Our study has an important implication for the data assimilation of GPS RO data. These days most of global NWP centres prefer bending angle to refractivity for data assimilation. Although there are good supporting reasons for the preference, this study finds that the assimilation of bending angle has drawbacks that are often disregarded in previous studies. It appears that VR is very promising as it reduces the EIV error and benefits from the PHD. It is straightforward to assimilate the refractivity

produced by VR. In order to take the full advantage of VR and to ensure consistency with the background, however, it will be desirable to incorporate the regularization into data assimilation methods. The recent version of COSMIC one-dimensional variational retrieval method (1D-Var), for instance, conducts VR on the fly prior to the actual retrieval. Alternatively, stand-alone VR that shares the background with data assimilation can be carried out as a RO data pre-processing step so as to reduce the computational complexity. An example is the 1D-Var+4D-Var approach for assimilation of precipitation-affected

microwave radiance at ECMWF (Bauer et al., 2006).

**Acknowledgements**

This material is based upon work supported by the National Science Foundation (NSF) under Cooperative Agreement No. AGS-1522830; and, by the National Aeronautics and Space Administration under Award No. NNX12AP89G issued through the Earth Science Division, Science Mission Directorate.

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

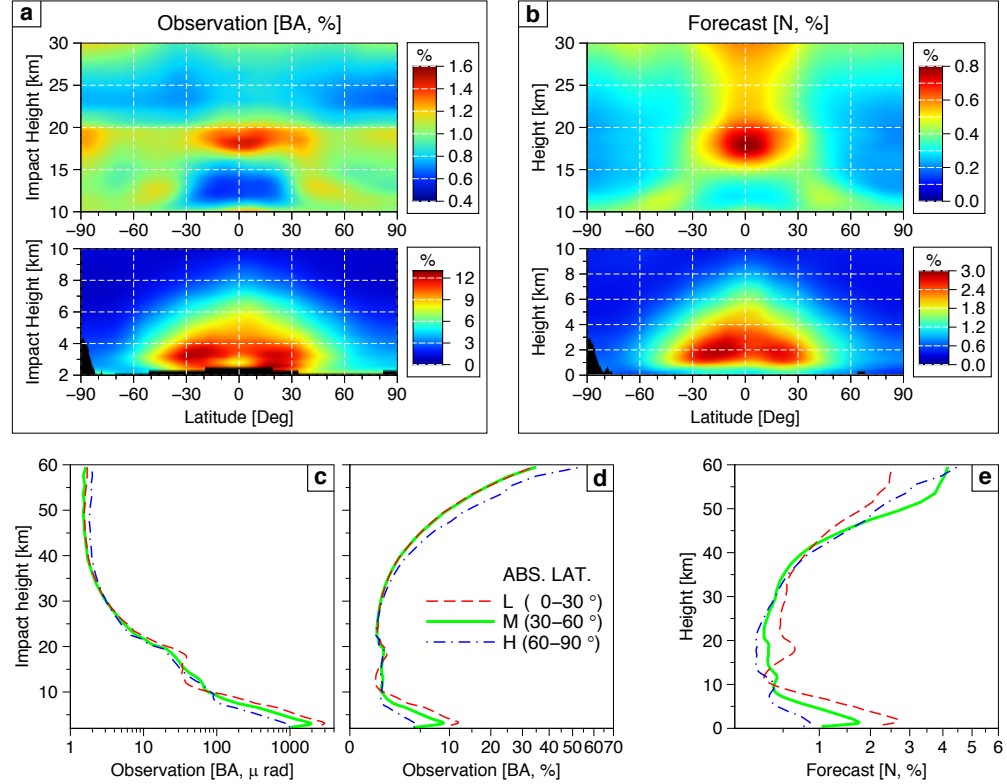

**Figure 1.** An example of diagonal elements of a) **R** and b) **B** diagnosed with HL86. Shown are the error standard deviations (%) of a) observed bending angle and b) forecast refractivity, averaged along the longitude and over the whole data period. The bending angle $\sigma_o$ averaged in three latitude bands (low latitude, red dashed; middle latitude, solid green; high latitude, blue dashed dot) is shown in the unit of c) $10^{-6}$ rad and d) %. e) is the same as in d) except for refractivity $\sigma_f$.

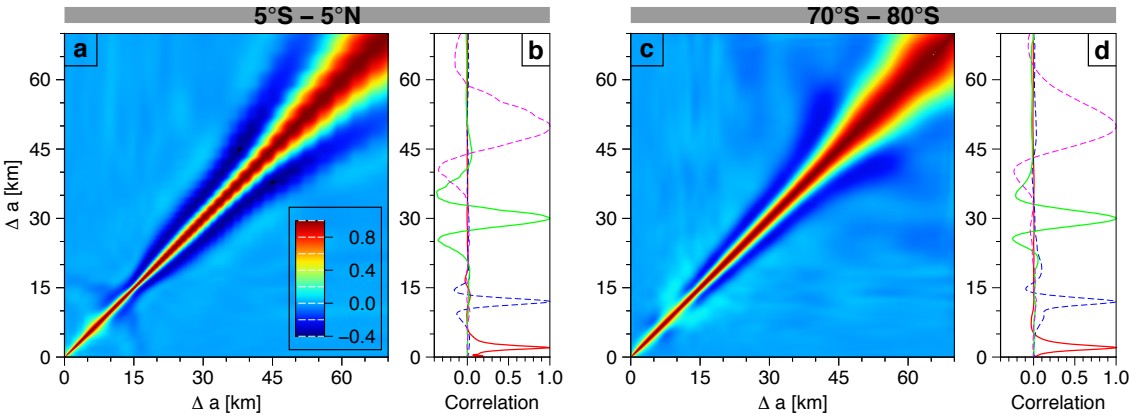

**Figure 2.** An example of forecast error correlation matrix made available through the NMC method. a) Forecast error correlation in a 10° latitude bin (5° S - 5° N), which is averaged along the longitude and during the months of July. The coordinate $\Delta a$ indicates the distance (km) in the refractional radius from the lower bound. b) Profiles of error correlation centred at four arbitrarily chosen heights. c) and d) are the same as in a) and b) except for in 70° S - 80° S bin.

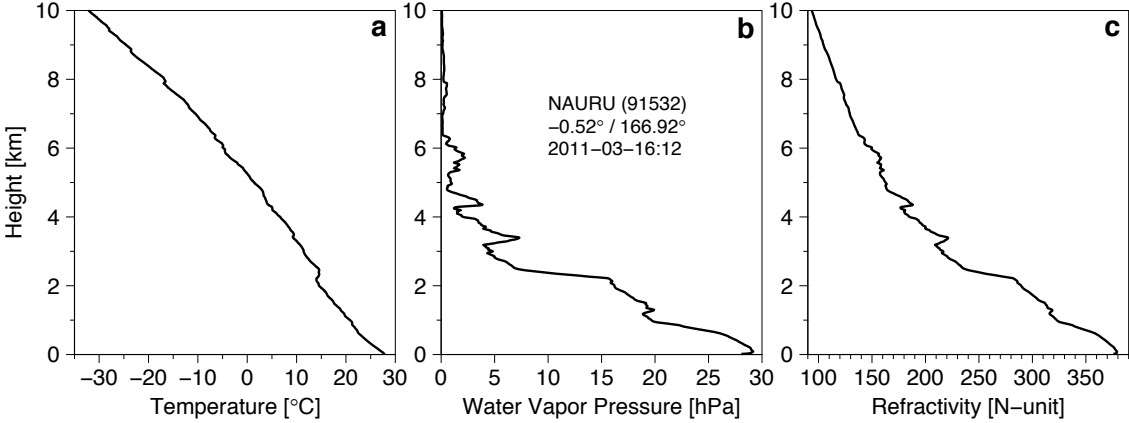

**Figure 3.** High-resolution radiosonde sounding at a tropical site in Nauru (0.52° S, 166.93° E) at 12:00 UT on 3 March 2011: a) temperature (° C), b) water vapor pressure (hPa), and c) refractivity (N-unit).

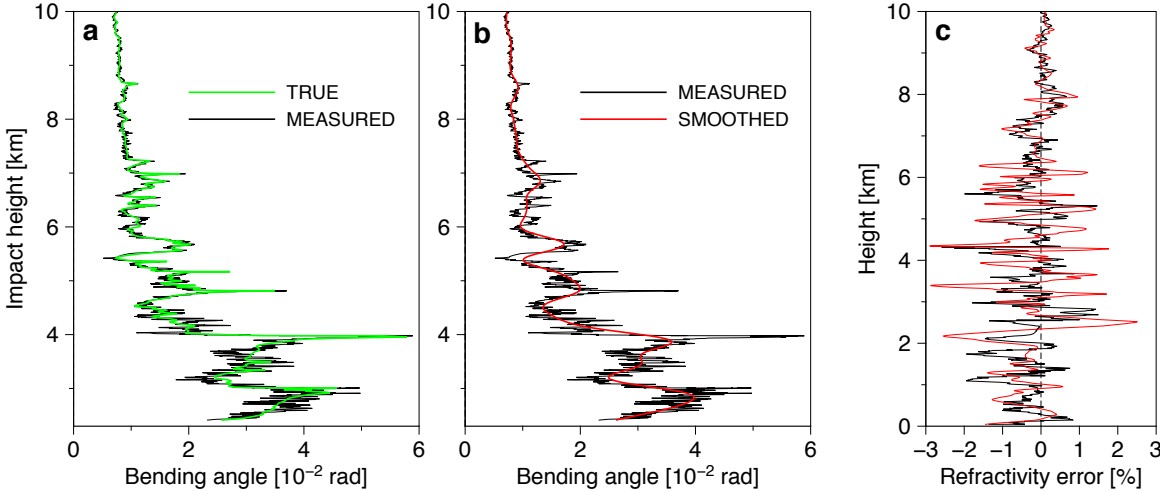

**Figure 4.** Simulated bending angles and their errors propagated into the refractivity via Abel inversion. a-b) Green line (denoted TRUE) indicates the bending angle modelled with the exact refractivity profile shown in Fig. 3c; black line (MEASURED) represents the "measured" bending angle for which assumed measurement errors are added to the perfect bending angle; red line (SMOOTHED) denotes a low-pass filtered version of the MEASURED. Note that the black line in b), which is the same as that in a), is duplicated to ease the comparison. c) Errors in the refractivity; the refractivity profiles are obtained from the bending angle of matching color via Abel inversion and the error is defined as the difference from the perfect refractivity, which is the one derived from the TRUE bending angle. See text for more details.

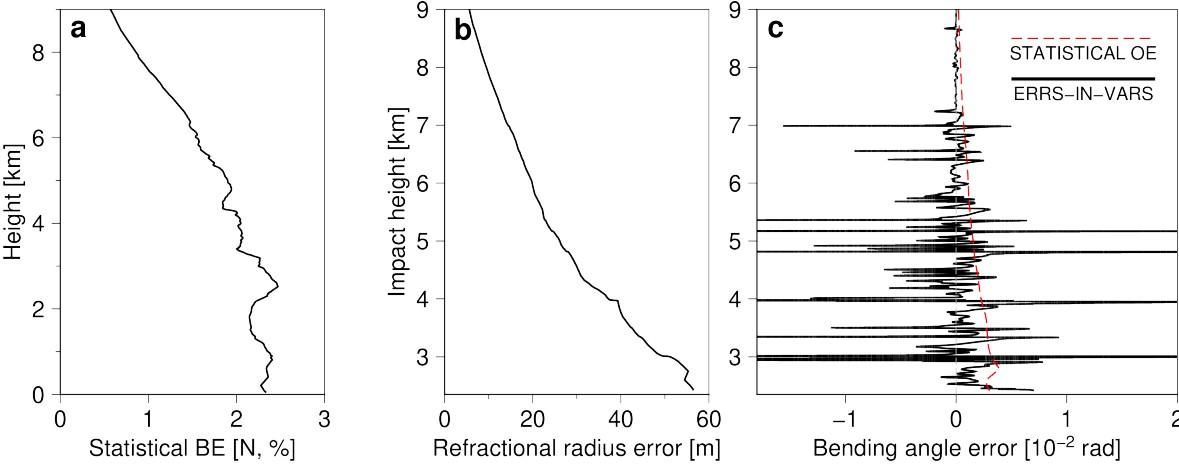

**Figure 5.** Errors involved in the test with a synthetic sounding: a) statistical estimate of model error in the refractivity at the time and location of the sounding; b) model error in the refractional radius resulting from that in the refractivity; c) statistical error estimate of bending angle observation (dashed red line) and the bending angle error due to the errors-in-variables problem (solid black), which is the modelling error of the synthetic bending angle stemming from the model error in the refractivity.

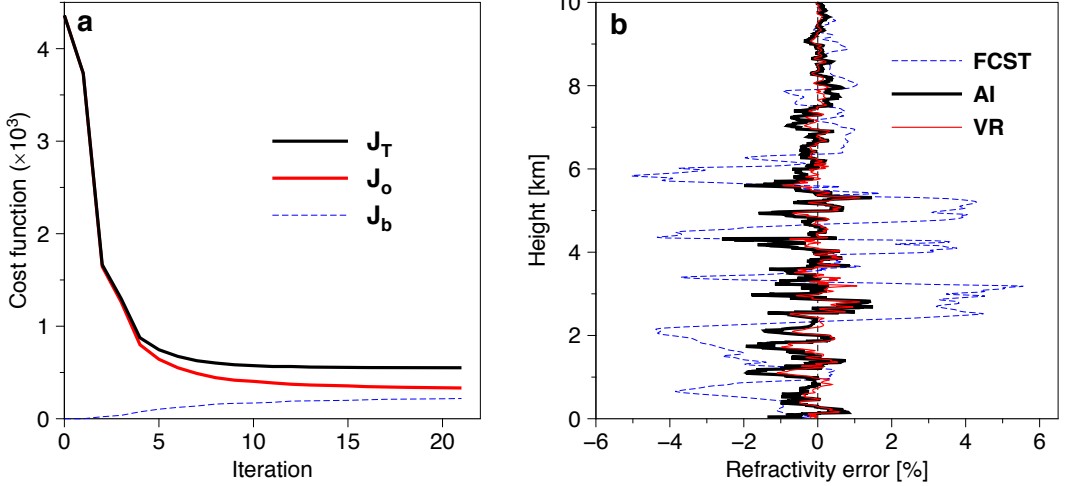

**Figure 6.** a) Change of the cost function with iteration in the variational regularization: observational term (denoted $J_o$, thick red line); background term ($J_b$, dashed blue); and, the total cost function ($J_T = J_o + J_b$, thick black). b) Refractivity error of the background (dashed blue), of the Abel inversion (thick black), and of the variational regularization (red).

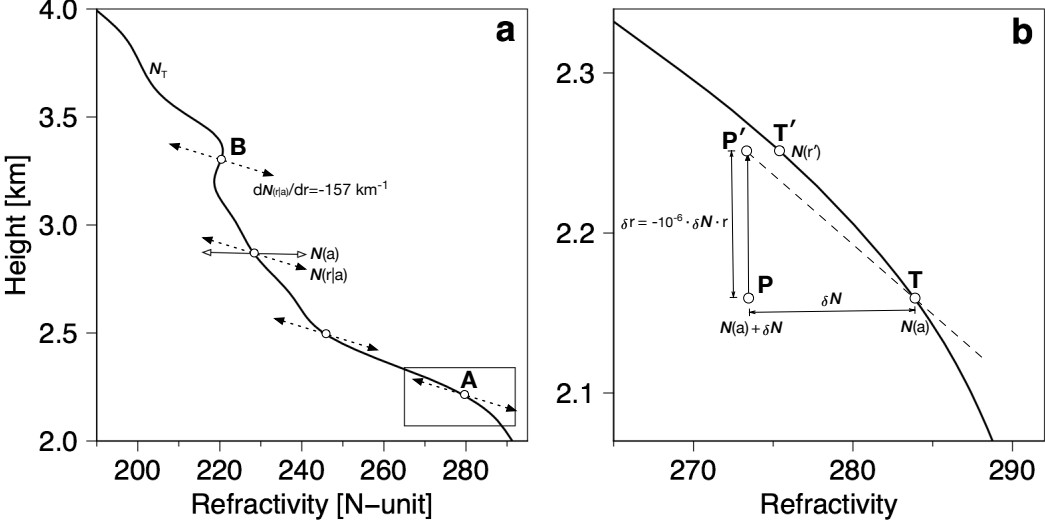

**Figure 7.** Illustration of solution's trajectory in the posterior height determination (PHD). a) The heavy solid curve represents the true refractivity sounding. The solid line with open-headed arrows around 2.8 km, denoted $N(a)$, indicates the actual trajectory (i.e., connected line of positions) of potential solutions for a given refractional radius (or true height). PHD relocates the solutions, making the trajectory to be interpreted as the dotted, slanted line next to $N(r|a)$. b) Magnified view of a relocation in the rectangular inset in a). In this example, the solution P is smaller than the true refractivity T and is thus relocated to P′, which is from the true radius $r$ to a higher location $r'$ since $r = xn^{-1}$. This leads the apparent error $\overline{|P'T'|}$ to be smaller than the true error $\overline{|PT|}$. The slope of $\overline{P'T}$ is constant, -157 km⁻¹, regardless of the true height or the size of true error as shown by the dotted lines in a). See text for more details.

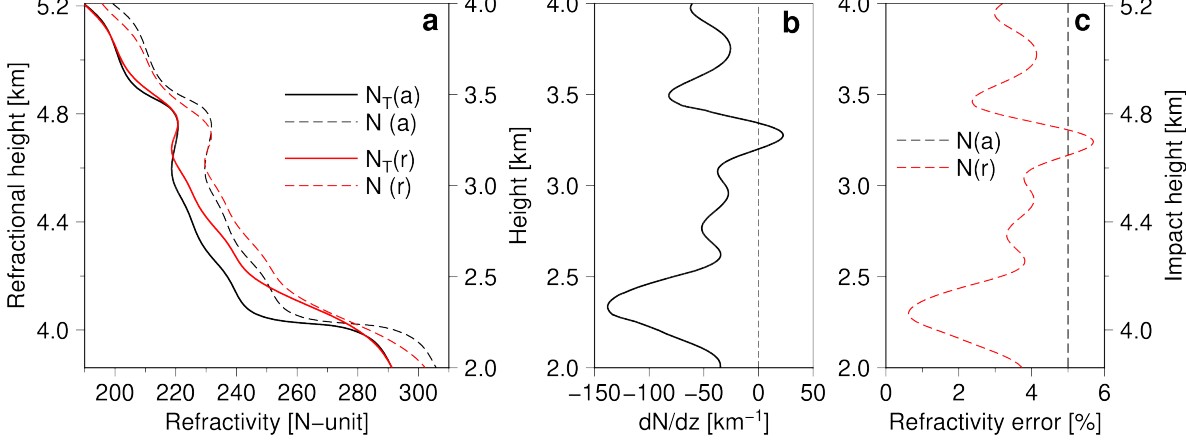

**Figure 8.** An illustrative example of true error versus apparent error. a) Black lines represent the true refractivity (solid) and a hypothetical solution (dashed) in the refractional radius coordinate. The hypothetical solution is set to be 5 % larger than the true refractivity in all heights. Red lines indicate those in the height coordinate, the true refractivity (solid) and the solution (dashed). b) The vertical gradient of the true refractivity. c) The true error (dashed black) and apparent error (dashed red), which are defined as the departure from the true refractivity in the same coordinate, i.e., dashed line minus solid line of the same color in a).

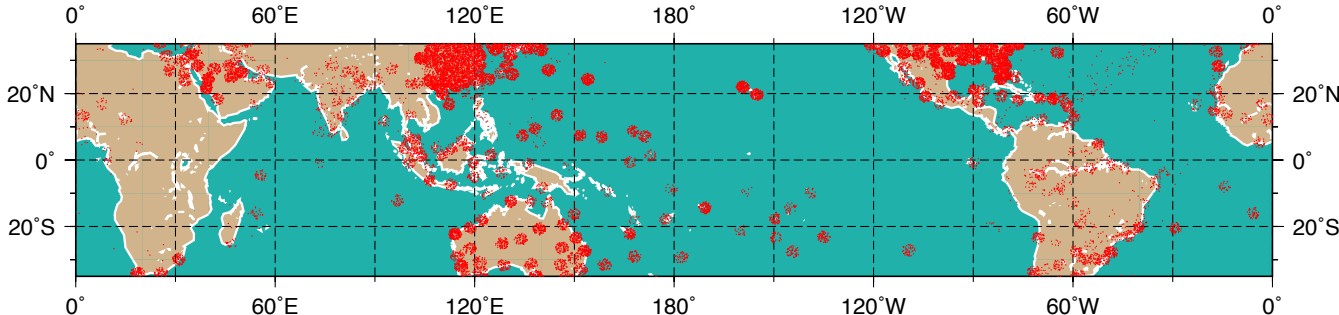

**Figure 9.** Location of COSMIC soundings (red dots) collocated with the operational radiosonde soundings in latitudes between 35° S and 35° N for the period from 17 February 2007 to 7 November 2010.

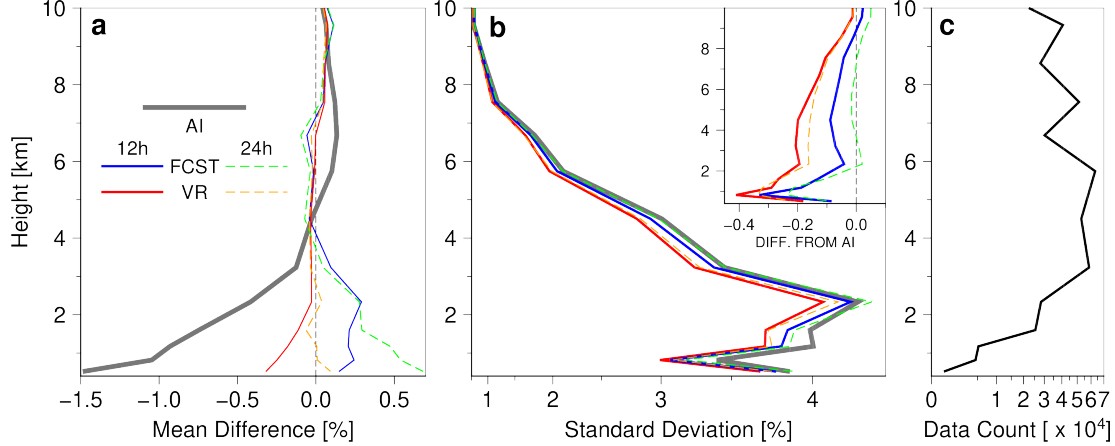

5    **Figure 10.** Comparison with collocated operational radiosonde data (ORD). a) Mean difference from ORD of CDAAC's refractivity derived via Abel inversion (thick dark grey line, denoted AI), ECMWF's 12h forecast (solid blue) and 24h forecast (dashed green), and VR-produced refractivity for which 12h (solid red) or 24h (dashed gold) forecast is used as the background. The unit used here is %, i.e., the percentage with respect to the mean value of ORD. b) Same as in a) except for the standard deviation. The inset shows the difference from AI in the standard deviation from ORD. c) Number of samples used in the comparison. Note that scales of the x-axis in b) and c) are not linear.

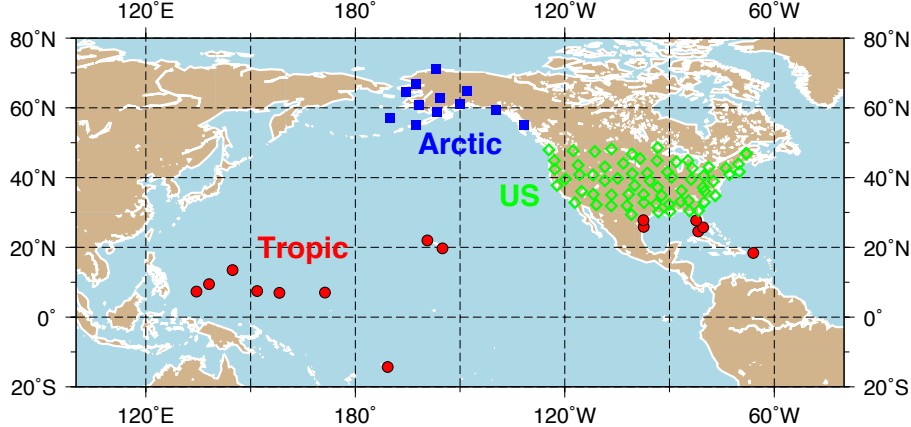

**Figure 11.** Location of US-owned HVRRD stations used in this study. The stations are classified into three latitudinal regions: the tropics (15 sites, red filled circles); US (64, green open diamonds); and, Arctic (Alaska) (13, blue filled squares).

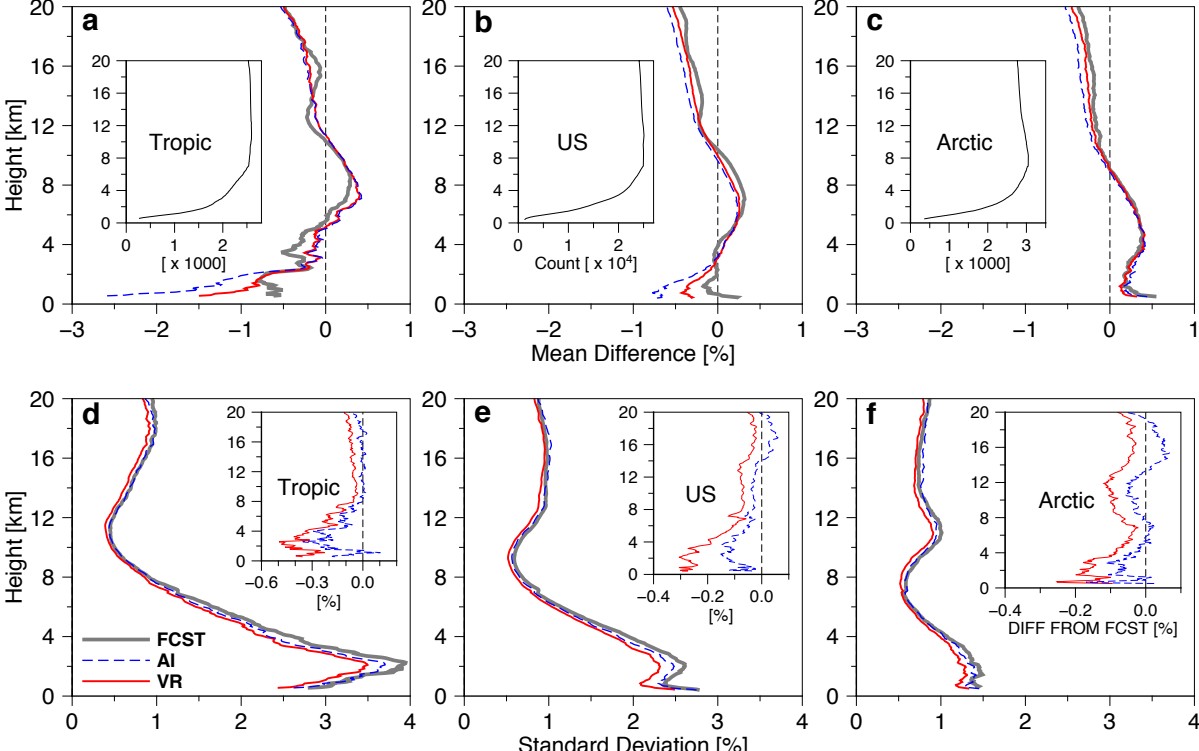

**Figure 12.** Comparison with collocated HVRRD. a-c) Mean difference (%)from HVRRD of CDAAC's refractivity derived via Abel inversion (dashed blue line, denoted AI), ECMWF's 24h forecast (heavy solid grey, FCST), and VR-produced refractivity (solid red) for which the FCST is used as the background. Shown are the statistics over a) the tropics, b) the US, and c) Alaska. The insets show the vertical data counts. d-f) The same as in a-c) except for the standard deviation. The insets show the differences from FCST in the standard deviation from HVRRD.