# Peer review of "A variational regularization of Abel transform for GPS radio occultation"

_Atmospheric Measurement Techniques, 2017_

## Referee Comment (RC1) · Anonymous Referee #1 · 24 Sep 2017

Overall, the paper is clear and concise and presents a case where using a variational technique can be used to improve the retrieval of a refractivity profile from GNSS-RO. The math and application of the technique all appear sound. However, the improved refractivity profile is acquired by the assimilation of ECMWF forecasted atmospheric profiles. This then poses a fundamental question as to the goal of GNSS-RO, is it to solely obtain the best refractivity profile, or to gain atmospheric state information such as temperature and moisture from the profile? If it's the latter there may then be an incestuous relationship where if the ECMWF model were to ingest these refractivity profiles, they would be ingesting forecast data from their own model. Does this technique then need to be applied for each NWP model independently, and the background error covariance calculated for the NWP model it is going to be applied to? Further, in

the technical description, I could use a little more detail on the computation particularly of the error covariances. The vertical coordinate is never discussed for these matrices and should be described. And though a reference is stated for the control-variable transform its application seems to merit a sentence or two. Lastly, the abstract of the paper itself never mentions that the regularization method will ingest ECMWF model forecast data, or more generically, NWP model forecast data, for input to the method. This is a key point, and obviously impactful on the final result and should be mentioned plainly in the abstract. Considering these points, revisions are required before publication can be considered, though minor revisions, they are fundamental and need to be addressed.

One last philosophical point, the paper should try to address the question as to what is the benefit of the final result of such a technique. The antagonist would say that in a full data assimilation system, would you acquire the same result assimilating bending angle or refractivity profile which utilizes the traditional AI approach, with appropriate observation error, and then also assimilating the ECMWF forecast model profiles? The benefit to the VR refractivity profiles is coming from the ECMWF model data, so if they are available why not just assimilate the ECMWF model data directly as proxy radiosondes? To address this concern, you could start by clearly stating that the goal or focus of this study is on creating the highest quality refractivity profile and what the benefits of such a dataset may be. Then follow up in the final summary and conclusions with a discussion about what may be the next steps in advancing this technique. It would seem that the logical extension would be to formulate a way to create a new forward operator for the bending angle profiles in the observation height coordinate which uses the NWP systems background (forecasts) to create an adjusted bending angle and PHD, and then subsequently transforms this back into innovations and Jacobians in the model space which can be used in the full solver minimization. It could be thought of as something similar to a 1D-Var step which would be embedded before passing information onto the main DA solver.

Typos and grammatical changes: Multiple times in the paper, data assimilation(s) is used. The final "s" is not needed as it can already be considered plural. One could use data assimilation methods/systems is you wanted to add another word, but it is perfectly adequate to leave this out. For example: Page 1, line 13: In contrast to variational data assimilation, VR holds . . . Page 1, line 17: . . . purposely corrupted synthetic sounding with a known true solution. Page 7, line 1: This differs from meteorological data assimilation of in-situ observations, where state variables are usually the same as those of the prediction model. — The original statement was not correct as currently the majority of observational data in meteorological data assimilation originates from satellite radiances which are not in state space, but need a forward operator similar to GNSS-RO. Please note my addition of "in-situ" but revise as you deem appropriate. Page 7, line 3: . . . the location of the state-vector elements is represented in relation . . . Page 7, line 4: For this reason, . . . Page 7, line 33: The method attempts to separate forecast and observation errors from the variance of $y - H(x)$, using the assumption that . . . Page 9, line 4: Question, does the sampling rate of 1 second correspond to 4 meters throughout the entire occultation or just in the lower troposphere. Please clarify, "4 meters though the depth of the occultation" or "4 meters in the lower troposphere" as appropriate. Page 10, line 8: On the other hand, regularization methods include the penalty term, which acts like a filtering and invokes a reverse . . . Page 11, line 10: Note, the total cost function in complex systems often does not always monotonically decrease. It can occur but often requires aggressive pre-conditioning to be applied and appropriate conjugate gradient descent methods chosen. You may want to preface that you have seen this behavior which can be attributed to the small order and general simplicity of the problem. Page 12, line 1: . . . Monte Carlo approach is larger than . . . Page 12, line 4: Data assimilation methods/systems are similar . . . Page 12, line 32: The slope is indeed the critical refractivity gradient, GC, . . . (This abbreviation was not defined, but subsequently used). Page 13, line 24: . . . actual RO events and compare these results with . . . Page 14, line 16: ... above the ECMWF model top up to a height of 2,000 km. Page 15, line 20: . . . does not always deviate discernibly from

ORD. Page 16, line 20: . . . RO and FCST is crucial to allow for VR to reduce the bias. Page 18, line 10: ... observations on average at the tropical HVRRD stations than the tropical ORD stations. Page 18, line 12: At the tropical HVRRD stations, RO observations are assumed to be more accurate, while the (simulated refractivities?) from the model background are less reliable. – I did not follow the use of "background sounding" so please be explicit here.

Figure 2: It would maybe help to have the "key" for the lines shown explicitly for figure 2a (true and measured), and figure 2b (smoothed and measured). Figure 3: I believe the x-axis in Figure 3a should be "Climatological B [N,%]" Figure 4: The purple and black lines are very hard to distinguish particularly in figure 4b. Figure 8: Similar to figure 4, the black and purple lines in figure 8b are very hard to distinguish.
* * *

---

## Referee Comment (RC2) · Anonymous Referee #2 · 5 Oct 2017

**Date: October 4, 2017**

Manuscript #:     amt-2017-228
Manuscript title: *A variational regularization of Abel transform for GPS radio occultation*

**Brief Summary of the Manuscript**
This manuscript proposes the variational regularization technique to estimate the atmospheric refractivity, given a-priori information from ECMWF. This proposed method is compared against radiosonde data and is applied in numerous GPS-RO retrievals. The advantage of this approach is that information about the error characteristics of the bending angle are also considered when estimating the refractivity, leading to better agreement with high resolution radiosondes at low altitudes. Despite the effort made, the manuscript in its current form is densely written, lacks motivation and objectivity, the methodology is not well placed within the context of the new approach, and the proposed technique is the 1-DVar that is routinely used in the retrieval of thermodynamic variables (except in this case is used to estimate the atmospheric refractivity). Sections of the paper are unnecessary and must be removed, because they provide textbook details that are beyond the scope of this study. The presentation of the results needs major re-writing with clear and concise explanation placed within the objective and motivation of this investigation. For examples, Section 2.1 can be reduced, Section 2.2 can be removed, Section 3.2 I do not understand its purpose, Section 3.3 the EIV issue is too detailed, Section 3.5 again too detailed and what purposed does it serve in this investigation, and so forth for the rest of the section. There are no references in the manuscript to back up the claims either. The proposed approach in the estimation of the atmospheric refractive index appears to introduce more effort and more error characterization than the usual AI approach and it also makes the refractivity more dependent on a-priori information. Given the above, I am afraid I cannot recommend this manuscript for publication in its present form.

**Major Comments:**
1) ***The introduction lacks motivation and objectivity and needs revising.*** The way the introduction is written does not establish the need for a new retrieval technique of the atmospheric refractivity and does not highlight the advantage(s) of the VR technique over the traditionally used Abel transform. Additionally, there are many unnecessary details on the VR that overwhelm the reader and overshadow any potential motivation and goals of this research. Proper referencing to the Abel transform limitations, other refractivity retrieval techniques, and the new VR method are also needed. What new information this VR could reveal over the Abel transform? Are the improvements within the Abel transform retrieval uncertainty and statistically significant?

2) ***The methodology lacks detailed explanation and is weak.*** The VR technique proposed in this study is practically a description of the 1-DVar assimilation approach, with the only difference being the assignment of the state variable (refractivity) onto the impact parameter space (unlike what is traditionally done in assimilation systems). Despite the fact that this may reduce the EIV problem, it is hardly a "breakthrough" as noted on Page 3 in Line 16 and the methodology lacks the following:
   a) Description of the error covariance matrix estimation
   b) How is the error covariance matrix for the forward model, **H(x)**, is defined in $\mathbf{R^{-1}}$?
   c) Page 9; Line 9: How is the "typical (climatic) size of the measurement" is defined?
   d) Page 9; Line 15: How sensitive is the bending angle error to the value of the parameter *L*?
   e) The forward modeled bending angle (**H(x)**) is estimated via Eq. (4), which is subject to the spherical symmetry approximation the author introduces as a limitation. Does this introduce additional retrieval errors to constraint through the VR?
   f) Page 7; Line 21: Since the initial guess matters, under what conditions can the author claim that the final solution does not depend on every detail on it? And what is meant by the "every detail"?
   g) How are the bending angle errors are defined in real GPS-RO measurements?
   h) What is the sensitivity of the VR to the initial guess from ECMWF?
   i) Page 16; Line 20: How does the "weighting" between observations and FCST is decided and

how sensitive this "weighting" is to the final solution?

   j) Page 18; Lines 9–11: Again, how is the "weighting" being defined in the VR technique?

   k) Page 19; Lines 6–7: Since the error carried forward into the refractivity, does not this imply that the forward model, **H(x)**, will also carry forward errors in the bending angle via the errors inherited through Eq. (4)?

   l) Page 20; Line 28: The estimation of the atmospheric refractivity appears to depend highly on the information provided by the initial guess, in this case ECMWF.

   m) Page 11; Line 10: How is the threshold of the cost function defined and how sensitive the final solution is to that threshold?

3) ***The results provide a marginal contribution to the state-of-the-art AI method of retrieving the atmospheric refractivity.*** Looking at Figure 4b, the refractivity error of the AI and of the VR methods falls within the retrieval uncertainty of the refractivity and is < 1.0% between the two. Is the difference between the black and the red lines statistically significant? The final solution also seems to depend heavily on the initial conditions (Page 20; Line 28), yet on Page 7 Line 21 the opposite is claimed, and the advantages of using the VR technique to estimate the refractivity are not clear through this presentation.

**Minor Comments:**
   a) **Line 5:** It should read: "In Radio Occultations (ROs), the refractivity is obtained from..."
   b) **Line 6:** It should read: "...AI is primarily susceptible to..." The only reason I want to see the word "primarily" in this sentence is because there are secondary mathematical issues that also limit the accuracy of the Abel transform.
   c) **Line 15:** How do you define "... known true solution... "
   d) **Lines 23-24:** It should read: "Knowledge of the refractive index vertical structure in the..."
   e) **Line 26:** Clarify what you mean by "flaws as well as signal components" and put a comma after the word "flaws"
   f) **Page 2; Line 1:** It should read: "... the atmosphere is one of the sources of..."
   g) **Page 2; Line 1:** I disagree that the non-spherical symmetry of the Earth's atmosphere qualifies as a bending angle measurement error. Bending angle is retrieved by the phase measurement; so I consider the non-spherical symmetry as a retrieval error, because the phase and amplitude of the GNSS signals measured at the LEO antenna are not affected by the spherical symmetry. Please, revise accordingly.
   h) **Page 2; Line 6:** Perhaps, change the word "measurement" with the word "retrieval"?
   i) **Page 2; Line 33-34:** It should read: "... to realistically estimate the uncertainties of RO measurement and *a-priori* and properly take them into account."
   j) **Page 3; Line 1:** Clarify what you mean by "reliable method"
   k) **Page 3; Line 1:** Remove the word "they"
   l) **Page 3; Line 2:** Add a recent reference regarding this statement.
   m) **Page 3; Line 16:** I would remove the word "breakthrough" and re-write the sentence as: "This necessitates the implementation of more rigorous techniques that can potentially improve the quality of the refractive index..."
   n) **Page 3; Lines 17-18:** It should read: "This study explores this possibility through the VR technique."
   o) **Page 3; Line 20:** It should read: "In Section 4, a real data validation..."
   p) **Page 3; Line 21:** It should read: "Section 5."
   q) **Page 3; Line 24:** It should read: "... and a receiver..."
   r) **Page 4; Line 5:** It should read: "... to be unstable."

s) **Page 4; Line 17:** It should read: "... operator in the RHS, in addition..."
t) **Page 4; Line 20:** Add a reference.
u) **Page 4; Line 27:** It should read: "... of the transmitted radiowave signal."
v) **Page 9; Line 18:** It should read: "Noise in the measured bending angle negatively affects the quality of the refractive index, unless properly mitigated."
w) **Page 15; Line 10:** Explain what you mean by "soundings that are dubious in quality are discarded".
x) **Page 16; Line 15:** Add a reference to back up this statement.
y) **Page 16; Line 28**: It should read: "... (especially in moisture)..."
z) **Page 17; Line 29**: It should read: "noise"
aa) **Page 18**; **Line 27**: Place a period instead of a comma after the word HVRRD.

---

## Author Comment (AC1) · 1 Jan 2018

Enclosed please find a zip file for my response to the Reviewer's comments.

Please also note the supplement to this comment:
https://www.atmos-meas-tech-discuss.net/amt-2017-228/amt-2017-228-AC1-supplement.zip

---

## Author Response (AR1)

**Response to Referee #1**

A variational regularization of Abel transform for GPS radio occultation

Tae-Kwon Wee  COSMIC project, University Corporation for Atmospheric Research P. O. Box 3000, Boulder, CO 80307, Boulder, Colorado

Overall, the paper is clear and concise and presents a case where using a variational technique can be used to improve the retrieval of a refractivity profile from GNSS-RO. The math and application of the technique all appear sound. However, the improved refractivity profile is acquired by the assimilation of ECMWF forecasted atmospheric profiles. This then poses a fundamental question as to the goal of GNSS-RO, is it to solely obtain the best refractivity profile, or to gain atmospheric state information such as temperature and moisture from the profile? If it's the latter there may then be an incestuous relationship where if the ECMWF model were to ingest these refractivity profiles, they would be ingesting forecast data from their own model. Does this technique then need to be applied for each NWP model independently, and the background error covariance calculated for the NWP model it is going to be applied to?

**Response)** I appreciate this observation. I admit that the incestuous relationship is unpleasant in theoretic perspective. However, it is difficult, if not impossible, to fully incorporate VR into data assimilation because of the EIV issue. Unfortunately, breaking the problem into two sequential steps (i.e., VR and then assimilating the retrieved refractivity) seems the only feasible option at the moment in order to avoid the EIV issue. The practical question is how significant the adverse effect could be compared to the benefit of VR. We have compared AI and VR by assimilating the refractivity data produced by them into WRF (Weather Research and Forecasting) DA. The DA experiments show that VR yields clear positive impacts on analysis and forecast over AI (manuscript is in preparation for separate publication). Based on the results, we strongly believe that the benefit of VR outweighs the incest effect. For instance, the 1D-Var+4D-Var approach for precipitation-affected microwave radiance at ECMWF (Bauer, P., Lopez, P., Salmond, D., Benedetti, A. and Moreau, E.: Implementation of 1D+4D-Var Assimilation of Microwave Radiances in Precipitation at ECMWF. I: 1D-Var, Q. J. R. Meteorol. Soc., 132, 2277–2306, 2006) is shown to be beneficial. Yes. It might be the best practice for individual NWP centers to apply VR independently using their own forecast and by constructing and using the background error covariance that is consistent with the forecast. This was stated in the last paragraph of section 5 (conclusions) of the original manuscript. By doing so, VR and data assimilation are consistent in the background and the potential contamination arising from external a priori can be avoided.

Further, in the technical description, I could use a little more detail on the computation particularly of the error covariances. The vertical coordinate is never discussed for these matrices and should be described. And though a reference is stated for the control-variable transform its application seems to merit a sentence or two. Lastly, the abstract of the paper itself never mentions that the regularization method will ingest ECMWF model forecast data, or more generically, NWP model forecast data, for input to the method. This is a key point, and obviously impactful on the final result and should be mentioned plainly in the abstract. Considering these points, revisions are required before publication can be considered, though minor revisions, they are fundamental and need to be addressed.

**Response)** Thank you so much for the valuable comments. All these are well taken in the revised manuscript.

One last philosophical point, the paper should try to address the question as to what is the benefit of the final result of such a technique. The antagonist would say that in a full data assimilation system, would you acquire the same result assimilating bending angle or refractivity profile which utilizes the traditional AI approach, with appropriate observation error, and then also assimilating the ECMWF forecast model profiles? The benefit to the VR refractivity profiles is coming from the ECMWF model data, so if they are available why not just assimilate the ECMWF model data directly as proxy radiosondes? To address this

A variational regularization of Abel transform for GPS radio occultation

Tae-Kwon Wee  COSMIC project, University Corporation for Atmospheric Research P. O. Box 3000, Boulder, CO 80307, Boulder, Colorado

**Brief Summary of the Manuscript**

This manuscript proposes the variational regularization technique to estimate the atmospheric refractivity, given a-priori information from ECMWF. This proposed method is compared against radiosonde data and is applied in numerous GPS-RO retrievals. The advantage of this approach is that information about the error characteristics of the bending angle are also considered when estimating the refractivity, leading to better agreement with high resolution radiosondes at low altitudes. Despite the effort made, the manuscript in its current form is densely written, lacks motivation and objectivity, the methodology is not well placed within the context of the new approach, and the proposed technique is the 1-DVar that is routinely used in the retrieval of thermodynamic variables (except in this case is used to estimate the atmospheric refractivity).

**Response)** I appreciate the reviewer's perspective. The reviewer's comments made me aware that the difference of the proposed approach from 1D-Var was not explained clearly in the original manuscript. The manuscript is now revised to articulate the point. The variational principle and method have been applied to a wide range of scientific problems and for various purposes. The variational technique is no more than a practical tool and using the technique in common does not make VR and 1D-Var the same. As an example, although 1D-Var and 3/4D-Var are essentially the same in the formulation, they are seldom considered to be the same. The specific purpose of the proposed method (regularization) is enhancing (noise-corrupted) RO observation. On the other hand, the main purpose of 1D-Var is finding the optimal combination of state (thermodynamic) variables. The focus of 1D-Var is thus on the behavior of the state variables and the relation among them. More importantly, the EIV issue prevents 1D-Var from yielding the same result with VR. Hence, as stated in the last paragraph of section 5, VR is applied separately prior to the 1D-Var step at COSMIC. In the author's perspective, the proposed regularization is fundamentally different from 1D-Var.

Sections of the paper are unnecessary and must be removed, because they provide textbook details that are beyond the scope of this study. The presentation of the results needs major re-writing with clear and concise explanation placed within the objective and motivation of this investigation. For examples, Section 2.1 can be reduced, Section 2.2 can be removed, Section 3.2 I do not understand its purpose, Section 3.3 the EIV issue is too detailed, Section 3.5 again too detailed and what purposed does it serve in this investigation, and so forth for the rest of the section. There are no references in the manuscript to back up the claims either. The proposed approach in the estimation of the atmospheric refractive index appears to introduce more effort and more error characterization than the usual AI approach and it also makes the refractivity more dependent on a-priori information. Given the above, I am afraid I cannot recommend this manuscript for publication in its present form.

**Response)** With all due respect to the reviewer, the structure of original manuscript was carefully designed after many considerations and I believe that each section is indispensable and presented as concisely as possible as explained in the following. The proposed method is on the regularization of Abel transform, described in **Section 2.1** and so a detailed description of the Abel transform pairs is crucial. Although a thorough description (rather than the brief introduction given in Section 2.1) could be more desirable, I wanted to keep it concise. Section 2.1 is also used to deliver the motivation and to raise the problem. **Section 2.2** is essential to introduce the concept of regularization and to explain the difference

(advantage) of the proposed method from (over) the classical regularization method. **Section 3.2** is important as it describes the practical application of Abel inversion in GPS RO. In actual RO data processing, Abel inversion cannot be applied to noisy "raw" bending angle. Therefore, the AI refractivity depends on details of the low-pass filtering (e.g., cutoff wavelength/frequency) applied to the input bending angle. Meanwhile, the error covariance matrices used in the proposed regularization act as a low-pass filter. Therefore, skeptics could say that the improvement shown by VR is nothing but the implicit filtering effect due to the error covariance matrices. Section 3.2 is purported to answer the question by presenting the limitation of filtering+AI (i.e., smoothing of bending angle, followed by AI). **Section 3.3** is to explain that 1D-Var is unable to reconstruct perfect refractivity out of perfect bending angle. As the reviewer stated, readers may consider VR to be similar with 1D-Var. The EIV issue poses the fundamental limitation on using bending angle in 1D-Var and data assimilation. The capability of addressing the issue is a unique advantage of VR and is the key that distinguishes VR from 1D-Var and data assimilation. The EIV issue has not been reported yet in the literature to the best of my knowledge and so I strongly believe that the issue deserves a detailed description. **Section 3.5** also deserves some space. These days, bending angle assimilation is considered to be de facto standard for RO data assimilation and the majority of global NWP prediction centers prefer bending angle to refractivity. The main reason is that bending angle is less processed compared to refractivity (e.g., Wee, T.-K. and Y.-H. Kuo, 2015: A perspective on the fundamental quality of GPS radio occultation data, Atmos. Meas. Tech., 8, 4281-4294, doi:10.5194/amt-8-4281-2015). However, the PHD is one of the reasons (in addition to model's limited top height and vertical resolution) that makes bending angle assimilation less favorable than refractivity assimilation, which is somewhat counterintuitive in view of error propagation because the additional processing (Abel inversion and preparation steps for the AI including the statistical optimization of bending angle in the stratosphere) introduces extra retrieval error. As the effect of PHD is so significant that it is compelling for NWP centers to reconsider the assimilation of bending angle. The PHD has not been published in peer-reviewed journals yet and substantiating the error reduction via PHD is an important contribution to RO science community. Because some of the findings presented in the manuscript are new, it is difficult to find peer-reviewed and citable previous studies. The proposed method is intended primarily for model-based applications such as 1D-Var and data assimilation. In those applications, RO data are eventually going to be combined with the model data. Therefore, the potential dependency is no concern. In addition, VR does not depend on the model much if RO observation is estimated to be trustworthy. For instance, VR virtually neglects the model above 3 km in Figs. 10a-b. VR respects the model when it reduces the total cost function effectively.

**Major Comments:**

*1) The introduction lacks motivation and objectivity and needs revising.* The way the introduction is written does not establish the need for a new retrieval technique of the atmospheric refractivity and does not highlight the advantage(s) of the VR technique over the traditionally used Abel transform. Additionally, there are many unnecessary details on the VR that overwhelm the reader and overshadow any potential motivation and goals of this research. Proper referencing to the Abel transform limitations, other refractivity retrieval techniques, and the new VR method are also needed. What new information this VR could reveal over the Abel transform? Are the improvements within the Abel transform retrieval uncertainty and statistically significant?

**Response)** The introduction is rewritten to address the reviewer's concerns. As explained above, the details on VR are given to describe the features that distinguish VR from AI and conventional data assimilation, also to present the originality of the work. Otherwise, readers might not comprehend the difference and benefit of VR. The detailed description is of course to emphasize the advantage(s) of the VR over the traditional AI. For spherically/radially asymmetric media, other inversion methods (e.g., radon transform, filtered backprojection, tomographic approach based on ray tracing, and so on) can be applied. For reconstruction of radially symmetric (i.e., 1-D) parameters, however, Abel transform

provides the mathematically exact solution for the given measurement. The question is of course whether the measurement is adequate to be used for Abel inversion. For instance, measurements in other fields/appications are often insufficient in the quality (e.g., accuracy, precision, completeness, coverage, and resolution). If the measurement permits, Abel inversion is undoubtedly the best approach. The proposed work does not attempt to develop new mathematical or analytical formulation/model for the inversion problem replacing Abel inversion. Instead, it uses Abel transforms and is indeed an overhaul of Abel inversion for imperfect observation. Thus, other refractivity retrieval techniques (that are not based on Abel transform) are largely irrelevant to this study. Again, the limitation of Abel inversion does not stem from the mathematical deficiency but from the imperfection of measurement, as repeatedly stated in the original manuscript. VR yields refractivity soundings of a higher quality by taking into account the measurement error. In the literature, to the best of the author's knowledge, there is no previous study that applied a variational regularization method that is similar to the approach proposed in this study to RO data. Although there are a fair number of studies that present variational regularizations, those are unrelated to RO and mostly at theoretic/conceptual level. The author could not find any relevant, and practical previous studies that are based on the formulation presented in this study. The author does not attempt to refute the efficacy of Abel inversion. On the contrary, it is acknowledged in the original manuscript that the refractivity retrieved via Abel inversion is of high quality, thanks to the superior quality of RO bending angle. Because of the very same reason, however, it is exceedingly difficult to further improve the data quality of refractivity. Meanwhile, the quality improvement (although it might not be sizable) is of crucial importance because RO refractivity is receiving ever-increasing attention from the science community. The sample numbers used in the validations are very large: 24,328 ORD (Fig. 8 in the original manuscript) and 30,796 HAVRRD (Fig. 10) soundings. The large number of samples (N) makes the standard error of the sampling distribution $SE \approx sqrt(2\frac{\sigma^2}{N})$ so small that the difference between AI and VR (especially the difference in the mean) is obviously significant without any need of significance tests. In addition, the difference between AI and VR in the standard deviation (e.g., Fig. 10 in the original manuscript) may look small because of the radiosonde error that is common to the standard deviations of AI and VR from the radiosonde data. That is, the standard deviations shown in Fig. 10 are: $\sigma_{AI-RS} = sqrt(\sigma_{AI}^2 + \sigma_{RS}^2)$ and $\sigma_{VR-RS} = sqrt(\sigma_{VR}^2 + \sigma_{RS}^2)$, where $\sigma_{AI}$, $\sigma_{VR}$, and $\sigma_{RS}$ indicate the random error of AI, VR, and radiosonde, respectively. The error estimation described in Section 2.3 shows that the radiosonde error is no smaller than RO observation error: $\sigma_{RS} \geq MAX(\sigma_{AI}, \sigma_{VR})$. When $\sigma_{AI}$ and $\sigma_{VR}$ are compared directly (instead of comparing $\sigma_{AI-RS}$ and $\sigma_{VR-RS}$), the difference between AI and VR appears much larger than what shown in Figs. 10d-f. This is only briefly mentioned in the original manuscript for the sake of conciseness but a manuscript is in preparation for separate publication. In summary, I strongly believe that the error reduction attained by VR is significantly large.

2) ***The methodology lacks detailed explanation and is weak***. The VR technique proposed in this study is practically a description of the 1-DVar assimilation approach, with the only difference being the assignment of the state variable (refractivity) onto the impact parameter space (unlike what is traditionally done in assimilation systems). Despite the fact that this may reduce the EIV problem, it is hardly a "breakthrough" as noted on Page 3 in Line 16 and the methodology lacks the following:

**Response)** The difference between VR and 1D-Var is now more clearly described in the revised manuscript.

a) Description of the error covariance matrix estimation

**Response)** Thanks for pointing these out. At the time that the original manuscript was written, the construction of error covariance matrices was considered as a straightforward, technical task and so detailed description was not given. I agree with the reviewer that additional description on the modeling of error covariance matrices can help readers understand the proposed method. A subsection is added to

the revision to describe the error covariance matrices.

b) How is the error covariance matrix for the forward model, $\mathbf{H(x)}$, is defined in $\mathbf{R}^{-1}$?

**Response)** The diagonal elements of $\mathbf{R}$ estimated through Hollingsworth-Lönnberg method include uncorrelated part of the observation operator error as well as the representativeness error. The non-instrument errors are not modeled separately.

c) Page 9; Line 9: How is the "typical (climatic) size of the measurement" is defined?

**Response)** The typical (climatic) measurement error means the measurement error estimated via the Hollingsworth-Lönnberg method and interpolated to the location and time of the RO sounding. To clarify that, it is reworded as statistical error.

d) Page 9; Line 15: How sensitive is the bending angle error to the value of the parameter $L$?

**Response)** The correlation of measurement error is determined by the parameter $L$, whereas the amplitude of measurement error is insensitive to the parameter. Low-pass filtering works great for purely random error, but it does not work well for strongly correlated error. Thus, longer $L$ is favorable for VR. In section 3.1, a short L is chosen on purpose not to overstate the effectiveness of VR over AI.

e) The forward modeled bending angle ($\mathbf{H(x)}$) is estimated via Eq. (4), which is subject to the spherical symmetry approximation the author introduces as a limitation. Does this introduce additional retrieval errors to constraint through the VR?

**Response)** Not only the "modeled" but also the "observed" bending angle is derived with the assumption of spherical symmetry in RO. Although VR does not use any explicit constraint that accounts for the horizontal inhomogeneity, the effect is reflected in the observation error.

f) Page 7; Line 21: Since the initial guess matters, under what conditions can the author claim that the final solution does not depend on every detail on it? And what is meant by the "every detail"?

**Response)** For instance, VR is close to AI and is insensitive to (distanced from) the first guess above 2 km in the Figs. 10a-b of the original manuscript. The initial guess helps VR more in the region that observed bending angle is less accurate. In addition, the initial guess is more respected if it reduces the total cost function effectively. The rapid reduction of $\mathbf{J}$ usually occurs when the initial guess can explain the modeled bending angle well in the observation space. Improper initial guesses, which are inconsistent with the observation, tend to fail to reduce the cost function effectually.

g) How are the bending angle errors are defined in real GPS-RO measurements?

**Response)** Without context, it is rather difficult to understand this question. In this study, the bending angle error is statistically estimated by applying the Hollingsworth-Lönnberg method to nearby pairs of RO-RO soundings collected for a long period. The error estimate is then interpolated to the time and location of individual soundings. Data providers generally do not provide the observation error and so it should be estimated by data users. While a measure of data uncertainty for few RO missions is available in a recent version of CDAAC data, it is not used in this study.

h) What is the sensitivity of the VR to the initial guess from ECMWF?

**Response)** The sensitivity depends on the assumed errors (error covariance matrices). More precisely, it depends on the trustworthiness of the error estimates. If the minimum of cost function can be found, the solution's sensitivity to the background can be approximated as: $\frac{\delta \mathbf{x}_a}{\delta \mathbf{x}_b} = 1 - (\mathbf{B}^{-1} + \mathbf{H}^T \mathbf{R}^{-1} \mathbf{H})^{-1} \mathbf{H}^T \mathbf{R}^{-1} \mathbf{H}$.

i) Page 16; Line 20: How does the "weighting" between observations and FCST is decided and how sensitive this "weighting" is to the final solution?

**Response)** It is determined by the error covariance matrices. For uncorrelated errors, the inverse of diagonal elements corresponds to the respective weighting. In general case, the weighting given to the observation relative to the background can be written as: $\mathbf{W} = \mathbf{BH}^T(\mathbf{R} + \mathbf{HBH}^T)^{-1}$; and, the relation of final solution $\mathbf{x}_a$ to background $\mathbf{x}_b$ is: $\mathbf{x}_a = \mathbf{x}_b + \mathbf{W}\{\mathbf{y}_o - H(\mathbf{x}_b)\}$. I believe that the sensitivity $(\frac{\delta \mathbf{x}_a}{\delta \mathbf{W}})$ is beyond the scope of this study and is more suitable for analytical, or ideal studies.

j) Page 18; Lines 9–11: Again, how is the "weighting" being defined in the VR technique?

**Response)** The weighting is the reciprocal of error variance, which varies with the impact height, latitude, and month of the year.

k) Page 19; Lines 6–7: Since the error carried forward into the refractivity, does not this imply that the forward model, $\mathbf{H}(\mathbf{x})$, will also carry forward errors in the bending angle via the errors inherited through Eq. (4)?

**Response)** Correct. In AI, the propagation of bending angle error cannot be controlled. In VR, however, the departure of model from observation, $\mathbf{y}_o - H(\mathbf{x})$, becomes smaller as the iteration proceeds, meaning that the refractivity error projected into the bending angle becomes smaller with the iteration. More importantly, the model is never assumed to be perfect and the mismodeling is included in $\mathbf{R}$. The error of observation operator (mostly discretization error) is concerning when the vertical data resolution is poor, e.g., NWP data of a crude vertical resolution. In VR, the discretization error is negligible as more than 800 vertical layers are used.

l) Page 20; Line 28: The estimation of the atmospheric refractivity appears to depend highly on the information provided by the initial guess, in this case ECMWF.

**Response)** There is some dependency in the mean below 2 km. However, as mentioned earlier, VR is largely insensitive to the initial guess above 2 km. The dependency in the lowest 2 km does not indicate the limitation of VR but the large uncertainty of RO data. For instance, the comparison to radiosonde data shows large negative bias in the tropics and middle latitudes, which is well known to exist. The fundamental cause of the dependency is the limitation of currently-available RO bending angles in the region. As indicated by the comparison to radiosonde, ECMWF forecasts are less biased. Therefore, the approach of VR to the forecasts in the mean makes perfect sense. Furthermore, VR always reduces random error variance compared to the forecasts. In order to reduce the dependency, the estimation of RO bending angle must be first improved. For instance, VR disregards the initial guess in Fig. 4b of the original manuscript.

m) Page 11; Line 10: How is the threshold of the cost function defined and how sensitive the final solution is to that threshold?

**Response)** The threshold is for the gradient of the cost function. The iteration terminates when the norm

of gradient becomes smaller than $10^{-8}$ of the initial value. The change to the solution is trivial once the norm falls below $10^{-3}$ of the initial value. The tight threshold is used to ensure the iteration not to terminate at a local minimum of the cost function.

**3) *The results provide a marginal contribution to the state-of-the-art AI method of retrieving the atmospheric refractivity.*** Looking at Figure 4b, the refractivity error of the AI and of the VR methods falls within the retrieval uncertainty of the refractivity and is < 1.0% between the two. Is the difference between the black and the red lines statistically significant? The final solution also seems to depend heavily on the initial conditions (Page 20; Line 28), yet on Page 7 Line 21 the opposite is claimed, and the advantages of using the VR technique to estimate the refractivity are not clear through this presentation.

**Response)** AI is a straightforward method rather than being "state of the art". The particular advantage of using the synthetic data is the known true solution, as explained in the original manuscript. Because of that, the comparison between AI and VR can be made without any ambiguity. In other words, every bit of the retrieval error is precisely known. Therefore, a single realization is sufficient and significance test is unnecessary. Moreover, assessment of statistical significance is irrelevant to this study as the experiment uses only a single realization. The error reduction achievable by VR depends on the character of the artificial measurement error and it varies with the spectral range. In the synthetic data case of this study, the bending angle error is close to random, containing high frequency oscillations. The Abel integral of high-frequency noises causes phase shifts in the refractivity, which in turn makes it impossible to recover some of high-frequency structures of the true refractivity. Because the true solution is known and complete reconstruction of true solution is impossible, it makes more sense to compare VR and AI in the relative error, $\frac{\varepsilon_{VR} - \varepsilon_{AI}}{\varepsilon_{AI}}$. As shown in Fig. 4b, VR is able to reduce 30-50 % of AI error. As explained earlier, the dependency of VR solution on the initial guess is determined by the quality of observation. In the synthetic data case, the solution does not depend on the background because the observation is unbiased (Fig. 2a). On the other hand, the real RO data have a significant negative bias and so the solution must approach the less biased background. AI works well in general because the overall quality of RO bending angle is superb. Because of that, it is also extremely difficult to further improve the quality of refractivity. The author has made enduring efforts to improve AI for the last 15 years, e.g., by tuning the degree of smoothing of bending angle and by using different discrete forms of the Abel inversion. However, the efforts were largely in vain not yielding any noticeable error reduction, which is understandable because, e.g., the smoothing applied to CDAAC's bending angle is polished; and, the data resolution of RO bending angle is very high, keeping the numerical error of discrete AI very small; and, radiosonde data are subject to considerable error, making the verification with it challenging. The error reduction through VR is significant, far exceeding those attainable by adjusting low-pass filtering and by using refined numerical schemes for the discrete AI.

**Minor Comments:**

**Line 5:** It should read: "In Radio Occultations (ROs), the refractivity is obtained from..."    **Response)** The RO "technique" is meant in the sentence. Change has been made.

**Line 6:** It should read: "...AI is primarily susceptible to..." The only reason I want to see the word "primarily" in this sentence is because there are secondary mathematical issues that also limit the accuracy of the Abel transform.

**Response)** Although it is uncertain which mathematical issues the reviewer is referring to, VR and AI are based on the same mathematical model because forward and inverse Abel transforms are exact mathematical inverse of each other. With all due respect to the reviewer, I am afraid the "primarily" does not add any new information.

**Line 15:** How do you define "... known true solution... "

**Response)** The true solution is known only for the synthetic observation described in section 3.2. The true solution is the perfect atmospheric sounding extracted from hypothetic true atmosphere. In the manuscript, the true solution is defined as the high-resolution radiosonde sounding as it is reported. The synthetic observation is simulated by applying the forward operator to the perfect solution and by adding assumed measurement error to the simulated (perfect) bending angle.

**Lines 23-24:** It should read: "Knowledge of the refractive index vertical structure in the..."

**Response)** Thanks for the suggestion. RO soundings provide horizontal structure as well when aggregated.

**Line 26:** Clarify what you mean by "flaws as well as signal components" and put a comma after the word "flaws"

**Response)** The sentence is rephrased.

**Page 2; Line 1:** It should read: "... the atmosphere is one of the sources of..."

**Response)** Change has been made.

**Page 2; Line 1:** I disagree that the non-spherical symmetry of the Earth's atmosphere qualifies as a bending angle measurement error. Bending angle is retrieved by the phase measurement; so I consider the non-spherical symmetry as a retrieval error, because the phase and amplitude of the GNSS signals measured at the LEO antenna are not affected by the spherical symmetry. Please, revise accordingly.

**Response)** I appreciate the reviewer's perspective. In the context of Abel inversion, the measurement refers to the input parameter, which is the bending angle. Therefore, the asymmetry is considered as measurement error. Two sentences are added to clarify that. Thank you very much for pointing this out.

**Page 2; Line 6:** Perhaps, change the word "measurement" with the word "retrieval"?

**Response)** As explained above, the gross bending angle error is treated as measurement error.

**Page 2; Line 33-34:** It should read: "... to realistically estimate the uncertainties of RO measurement and *a-priori* and properly take them into account."

**Response)** In accordance with the reviewer's suggestion, positions of the adverbs are changed.

**Page 3; Line 1:** Clarify what you mean by "reliable method"

**Response)** The sentence is now reworded to improve the clarity. Thanks.

**Page 3; Line 1:** Remove the word "they"

**Response)** Removed.

**Page 3; Line 2:** Add a recent reference regarding this statement.

**Response)** The Tikhonov regularization is now specified, which is described in detail in section 2.2. There, readers will find that the Tikhonov regularization does not make use of data uncertainties. While the Tikhonov regularization has been used in broad applications, it is hard to find recent publications relevant to GPS RO. I believe that the statement made in the sentence qualifies as common knowledge.

**Page 3; Line 16:** I would remove the word "breakthrough" and re-write the sentence as: "This necessitates the implementation of more rigorous techniques that can potentially improve the quality of the refractive index..."

**Response)** The sentence is reworded reflecting the reviewer's suggestion.

**Page 3; Lines 17-18:** It should read: "This study explores this possibility through the VR technique."

**Response)** The preposition is change from "with" to "through" as the reviewer suggested.

**Page 3; Line 20:** It should read: "In Section 4, a real data validation..."

**Response)** Done.

**Page 3; Line 21:** It should read: "Section 5."

**Response)** It is required by AMT guideline (https://www.atmospheric-measurement-techniques.net/for_authors/manuscript_preparation.html): The abbreviation "Sect." should be used when it appears in running text and should be followed by a number unless it comes at the beginning of a sentence.

**Page 3; Line 24:** It should read: "... and a receiver..."

**Response)** Done. Thank you very much for pointing this out.

**Page 4; Line 5:** It should read: "... to be unstable."

**Response)** Corrected.

**Page 4; Line 17:** It should read: "... operator in the RHS, in addition..."

**Response)** Corrected. Thank you.

**Page 4; Line 20:** Add a reference.

**Response)** Done.

**Page 4; Line 27:** It should read: "... of the transmitted radiowave signal."

**Response)** The SNR mentioned in the sentence is different from the phase SNR. Bending angle noise depends on many other factors.

**Page 9; Line 18:** It should read: "Noise in the measured bending angle negatively affects the quality of the refractive index, unless properly mitigated."

**Response)** Corrected.

**Page 15; Line 10:** Explain what you mean by "soundings that are dubious in quality are discarded".

**Response)** The sentence is revised to improve the clarity.

**Page 16; Line 15:** Add a reference to back up this statement.

**Response)** A reference is now added although the statement was meant to be a speculation.

**Page 16; Line 28**: It should read: "... (especially in moisture)..."

**Response)** Done.

**Page 17; Line 29**: It should read: "noise"

**Response)** Corrected.

**Page 18**; **Line 27**: Place a period instead of a comma after the word HVRRD.

**Response)** Change is made. Thank you very much.

concern, you could start by clearly stating that the goal or focus of this study is on creating the highest quality refractivity profile and what the benefits of such a dataset may be. Then follow up in the final summary and conclusions with a discussion about what may be the next steps in advancing this technique. It would seem that the logical extension would be to formulate a way to create a new forward operator for the bending angle profiles in the observation height coordinate which uses the NWP systems background (forecasts) to create an adjusted bending angle and PHD, and then subsequently transforms this back into innovations and Jacobians in the model space which can be used in the full solver minimization. It could be thought of as something similar to a 1D-Var step which would be embedded before passing information onto the main DA solver.

**Response)** The manuscript has been significantly revised to follow the reviewer's suggestions. With all due respect, I believe that many of these points are already explained in the original manuscript, but not clearly enough. The introduction is completely rewritten to articulate these points. In case of AI-produced refractivity, it is impossible for data assimilation to undo the vertical propagation of bending angle error. For bending angle data assimilation, the EIV problem makes it impossible to acquire the same result with AI. For instance, the data assimilation is unable to "retrieve" perfect refractivity out of error-free bending angle unless the provided background refractivity is initially perfect. Therefore, it is impossible to yield the same result with VR by assimilating bending angle or refractivity profile even with assigning proper observation error. The practical constraints of NWP models (limited top height and vertical resolution) are an additional issue. Note that RO data processing typically uses a significantly higher data resolution (number of layers in the order of thousands or more) and top height (2,000 km as described in the manuscript) for AI. Numerous Observing System Experiments have shown the positive impact of RO data (even for AI-produced RO refractivity), which cannot be attained by assimilating forecast profiles. In addition, the error estimation described in section 2.3 of the manuscript shows that AI is superior to ECMWF forecast in the tropospheric refractivity. It means that the information in VR mainly comes from RO bending angle rather than the forecast. Although it is not possible to eliminate the forecast influence completely, VR-produced refractivity is certainly better than the forecast in the quality.

Typos and grammatical changes: Multiple times in the paper, data assimilation(s) is used. The final "s" is not needed as it can already be considered plural. One could use data assimilation methods/systems is you wanted to add another word, but it is perfectly adequate to leave this out. For example: Page 1, line 13: In contrast to variational data assimilation, VR holds . . .

**Response)** Done. Thank you very much for pointing this out.

Page 1, line 17: . . . purposely corrupted synthetic sounding with a known true solution.

**Response)** Corrected.

Page 7, line 1: This differs from meteorological data assimilation of in-situ observations, where state variables are usually the same as those of the prediction model. — The original statement was not correct as currently the majority of observational data in meteorological data assimilation originates from satellite radiances which are not in state space, but need a forward operator similar to GNSS-RO. Please note my addition of "in-situ" but revise as you deem appropriate.

**Response)** The above-mentioned sentence does not state that the observed variables are of the same type with model (state) variables. The sentence explains that the state/control variable of VR (refractivity), instead of observed variable, is different from model variables. The control (state) variables in the data assimilation of satellite radiances are still a subset of model prognostic variables (temperature, moisture, surface pressure, winds, …) and the forward operator (radiative transfer model) maps the state variables into the observation space. Assimilation of indirect variables does not necessitate any change of the state vector (i.e., constituent elements or structure).

Page 7, line 3: . . . the location of the state-vector elements is represented in relation . . .

**Response)** Changed. Thanks for this excellent suggestion.

Page 7, line 4: For this reason, . . .

**Response)** Done.

Page 7, line 33: The method attempts to separate forecast and observation errors from the variance of y – H(x), using the assumption that . . .

**Response)** The method is able to separate the errors.

Page 9, line 4: Question, does the sampling rate of 1 second correspond to 4 meters throughout the entire occultation or just in the lower troposphere. Please clarify, "4 meters though the depth of the occultation" or "4 meters in the lower troposphere" as appropriate.

**Response)** The sampling rate is meant for radiosonde data. The balloon ascent rate is quite variable and depends on many factors (e.g., atmospheric stratification and vertical air motion). The ascent rate shows large sounding-to-sounding variability but does seem to have strong height dependency. After rechecking the radiosonde dataset, the ascent rate is now changed to "about 5 meters" and "on average" is added to indicate the high variability.

Page 10, line 8: On the other hand, regularization methods include the penalty term, which acts like a filtering and invokes a reverse . . .

**Response)** Changed. Thanks for this good suggestion.

Page 11, line 10: Note, the total cost function in complex systems often does not always monotonically decrease. It can occur but often requires aggressive pre-conditioning to be applied and appropriate conjugate gradient descent methods chosen. You may want to preface that you have seen this behavior which can be attributed to the small order and general simplicity of the problem.

**Response)** I agree that it is more difficult for minimization algorithms to find the steepest descent direction in large-scale problems. Subsequently, the cost function may not decrease as fast as shown in Fig. 4. However, the total cost function must decrease monotonically with iteration. Otherwise, the iteration simply terminates (although the cost function may increase in trial steps between iterations, the trial steps are not considered as a successful iteration unless a reduced cost function can be found). The sentence is rephrased to imply that the convergence speed can be dependent on the problem size. Your suggestion is greatly appreciated.

Page 12, line 1: . . . Monte Carlo approach is larger than . . .

**Response)** Corrected. Thank you.

Page 12, line 4: Data assimilation methods/systems are ORD.

**Response)** Changed.

Page 16, line 20: . . . RO and FCST is crucial to allow for VR to reduce the bias.

**Response)** As suggested, "though" is now deleted.

Page 18, line 10: ... observations on average at the tropical HVRRD stations than the tropical ORD stations.

**Response)** Thank you; "tropical" is added.

Page 18, line 12: At the tropical HVRRD stations, RO observations are assumed to be more accurate, while the (simulated refractivities?) from the model background are less reliable. – I did not follow the use of "background sounding" so please be explicit here.

**Response)** Your comment is well taken. The sentence has been rephrased to make it easy to understand. Thanks for pointing this out.

Figure 2: It would maybe help to have the "key" for the lines shown explicitly for figure 2a (true and measured), and figure 2b (smoothed and measured).

**Response)** Figure 2a-b are modified as your kind suggestion.

Figure 3: I believe the x-axis in Figure 3a should be "Climatological B [N,%]"

**Response)** Correct. Decided to use "statistical". Thank you.

Figure 4: The purple and black lines are very hard to distinguish particularly in figure 4b.

**Response)** The blue (looks like purple somehow) line is now dashed. Thank you for this great suggestion.

Figure 8: Similar to figure 4, the black and purple lines in figure 8b are very hard to distinguish.

**Response)** The Figure is revised to improve the clarity.

Page 12, line 32: The slope is indeed the critical refractivity gradient, GC, . . . (This abbreviation was not defined, but subsequently used).

**Response)** It is now defined explicitly.

Page 13, line 24: . . . actual RO events and compare these results with . . .

**Response)** The sentence is reworded. Thank you.

Page 14, line 16: ... above the ECMWF model top up to a height of 2,000 km.

**Response)** Changed as your suggestion.

Page 15, line 20: . . . does not always deviate discernibly from

**Response)** Done. Thank you.

[revised manuscript text omitted]

Formatted ... [66]
Formatted ... [67]
Formatted ... [68]
Formatted ... [70]
Formatted ... [69]
Formatted ... [71]
Formatted ... [72]
Formatted ... [74]
Formatted ... [75]
Formatted ... [76]
Formatted ... [77]
Formatted ... [78]
Formatted ... [73]
Formatted ... [79]
Formatted ... [80]
Formatted ... [81]
Formatted ... [82]
Formatted ... [83]
Formatted ... [84]
Formatted ... [85]
Formatted ... [86]
Formatted ... [87]
Formatted ... [88]
Formatted ... [89]
Formatted ... [90]
Formatted ... [91]
Formatted ... [92]
... [93]
Formatted ... [94]
Formatted ... [95]
Formatted ... [96]
Formatted ... [97]
Formatted ... [98]

[revised manuscript text omitted]

Don't adjust right indent when grid is defined, No widow/orphan control, Don't adjust space between Latin and Asian text, Don't adjust space between Asian text and numbers

| Page 8: [9] Deleted | Revision | 1/1/18 1:55:00 PM |

$S$ shown above suppresses such high-frequency modes effectively, it is difficult to choose a

| Page 8: [10] Deleted | Revision | 1/1/18 1:55:00 PM |

. The difficulty is not different from what one may encounter designing an optimal filter. Moreover, the particular $S$ does not account for any height dependency of

**Page 9: [11] Deleted**           **Revision**           **1/1/18 1:55:00 PM**

spatiotemporal variations because neither $S$ nor $F$ does. However, $\xi$ is typically a constant (i.e., scalar) and it does not have any structural information inside regarding the spatiotemporal variations.

Given that only noisy measurements are available in the real world,

**Page 9: [12] Deleted**           **Revision**           **1/1/18 1:55:00 PM**

An obvious distinction from TR in the formulation is that VR makes use of the error covariance matrices that conveys the prior information about the uncertainty of $\mathbf{y}_o$ and $\mathbf{x}_b$. In our study, $\mathbf{B}$ and $\mathbf{R}$ are built for each month on a $5° \times 5°$ latitude-longitude grid. This allows VR to account for spatiotemporal variation of the data uncertainty. Specifically, the normalization of $\mathbf{x} - \mathbf{x}_b$ and $\mathbf{y}_o - H(\mathbf{x})$ with their respective error, which is the square root of the diagonal elements of $\mathbf{B}$ and $\mathbf{R}$, eliminates the dependency of RHS terms on height, latitude, season, and so forth. In addition, the off-diagonal elements of the matrices permit VR to take into account spatial error correlations, which is essential for the background term

**Page 9: [13] Deleted**           **Revision**           **1/1/18 1:55:00 PM**

The forward observation operator used here is Eq. (4), which maps the state variable (refractive index) onto the observation (bending angle). It must be emphasized that the state variable (control variable to be precise) of VR resides in the impact parameter space where the observation exists, as can be conjectured from Eq. (4). This differs from meteorological

**Page 9: [14] Deleted**           **Revision**           **1/1/18 1:55:00 PM**

assimilations the state variables of which are usually the same as those of the prediction model: temperature, moisture, pressure, wind, and so on. In the data assimilations

**Page 10: [15] Deleted**           **Revision**           **1/1/18 1:55:00 PM**

, relying on the never-perfect model's refractive index. Hence, the modelled impact parameter

| Page 10: [15] Deleted | Revision | 1/1/18 1:55:00 PM |
|---|---|---|

, relying on the never-perfect model's refractive index. Hence, the modelled impact parameter

| Page 10: [16] Deleted | Revision | 1/1/18 1:55:00 PM |
|---|---|---|

The posterior height determination

| Page 10: [17] Deleted | Revision | 1/1/18 1:55:00 PM |
|---|---|---|

VR,

| Page 10: [18] Deleted | Revision | 1/1/18 1:55:00 PM |
|---|---|---|

A

| Page 10: [18] Deleted | Revision | 1/1/18 1:55:00 PM |
|---|---|---|

A

| Page 10: [18] Deleted | Revision | 1/1/18 1:55:00 PM |
|---|---|---|

A

| Page 10: [18] Deleted | Revision | 1/1/18 1:55:00 PM |
|---|---|---|

A

| Page 10: [18] Deleted | Revision | 1/1/18 1:55:00 PM |
|---|---|---|

A

| Page 10: [18] Deleted | Revision | 1/1/18 1:55:00 PM |
|---|---|---|

A

| Page 10: [18] Deleted | Revision | 1/1/18 1:55:00 PM |
|---|---|---|

A

| Page 10: [19] Deleted | Revision | 1/1/18 1:55:00 PM |
|---|---|---|

forecast and

| Page 10: [19] Deleted | Revision | 1/1/18 1:55:00 PM |
|---|---|---|

forecast and

| Page 10: [19] Deleted | Revision | 1/1/18 1:55:00 PM |
|---|---|---|

forecast and

| Page 10: [20] Deleted | Revision | 1/1/18 1:55:00 PM |
|---|---|---|

(

| Page 10: [20] Deleted | Revision | 1/1/18 1:55:00 PM |
|---|---|---|

(

| Page 10: [21] Deleted | Revision | 1/1/18 1:55:00 PM |
|---|---|---|

and the forecasts of the European Centre for Medium-Range Weather Forecasts (ECMWF) interpolated to the location of RO soundings. The method separates forecast and observation errors from the variance of $\mathbf{y}_o - H(\mathbf{x})$, under

| Page 10: [22] Formatted | Revision | 1/1/18 1:55:00 PM |
|---|---|---|

Font color: R,G,B (26,23,24)

| Page 10: [22] Formatted | Revision | 1/1/18 1:55:00 PM |
|---|---|---|

Font color: R,G,B (26,23,24)

| Page 10: [22] Formatted | Revision | 1/1/18 1:55:00 PM |
|---|---|---|

Font color: R,G,B (26,23,24)

| Page 15: [23] Deleted | Revision | 1/1/18 1:55:00 PM |
|---|---|---|

We then generate

| Page 16: [24] Deleted | Revision | 1/1/18 1:55:00 PM |

on the other hand the penalty term, which

| Page 16: [24] Deleted | Revision | 1/1/18 1:55:00 PM |

on the other hand the penalty term, which

| Page 16: [24] Deleted | Revision | 1/1/18 1:55:00 PM |

on the other hand the penalty term, which

| Page 16: [25] Deleted | Revision | 1/1/18 1:55:00 PM |

none

| Page 16: [25] Deleted | Revision | 1/1/18 1:55:00 PM |

none

| Page 16: [25] Deleted | Revision | 1/1/18 1:55:00 PM |

none

| Page 16: [25] Deleted | Revision | 1/1/18 1:55:00 PM |

none

| Page 16: [25] Deleted | Revision | 1/1/18 1:55:00 PM |

none

| Page 16: [25] Deleted | Revision | 1/1/18 1:55:00 PM |

none

| Page 16: [25] Deleted | Revision | 1/1/18 1:55:00 PM |

none

| Page 16: [25] Deleted | Revision | 1/1/18 1:55:00 PM |

none

| Page 16: [25] Deleted | Revision | 1/1/18 1:55:00 PM |

none

| Page 16: [25] Deleted | Revision | 1/1/18 1:55:00 PM |

none

| Page 16: [25] Deleted | Revision | 1/1/18 1:55:00 PM |

none

| **Page 16: [25] Deleted** | **Revision** | **1/1/18 1:55:00 PM** |
|---|---|---|

none

| **Page 16: [25] Deleted** | **Revision** | **1/1/18 1:55:00 PM** |
|---|---|---|

none

| **Page 16: [25] Deleted** | **Revision** | **1/1/18 1:55:00 PM** |
|---|---|---|

none

| **Page 16: [26] Deleted** | **Revision** | **1/1/18 1:55:00 PM** |
|---|---|---|

is

| **Page 16: [26] Deleted** | **Revision** | **1/1/18 1:55:00 PM** |
|---|---|---|

is

| **Page 16: [26] Deleted** | **Revision** | **1/1/18 1:55:00 PM** |
|---|---|---|

is

| **Page 16: [27] Deleted** | **Revision** | **1/1/18 1:55:00 PM** |
|---|---|---|

3b;

| **Page 16: [27] Deleted** | **Revision** | **1/1/18 1:55:00 PM** |
|---|---|---|

3b;

| **Page 16: [28] Deleted** | **Revision** | **1/1/18 1:55:00 PM** |
|---|---|---|

Around those heights a closer fit to

| **Page 16: [28] Deleted** | **Revision** | **1/1/18 1:55:00 PM** |
|---|---|---|

Around those heights a closer fit to

| **Page 16: [28] Deleted** | **Revision** | **1/1/18 1:55:00 PM** |
|---|---|---|

Around those heights a closer fit to

| **Page 16: [28] Deleted** | **Revision** | **1/1/18 1:55:00 PM** |
|---|---|---|

Around those heights a closer fit to

| **Page 16: [28] Deleted** | **Revision** | **1/1/18 1:55:00 PM** |
|---|---|---|

Around those heights a closer fit to

| **Page 16: [29] Deleted** | **Revision** | **1/1/18 1:55:00 PM** |
|---|---|---|

if

| **Page 16: [29] Deleted** | **Revision** | **1/1/18 1:55:00 PM** |
|---|---|---|

if

| **Page 16: [29] Deleted** | **Revision** | **1/1/18 1:55:00 PM** |
|---|---|---|

if

| Page 16: [29] Deleted | Revision | 1/1/18 1:55:00 PM |

if

| Page 16: [29] Deleted | Revision | 1/1/18 1:55:00 PM |

if

| Page 16: [29] Deleted | Revision | 1/1/18 1:55:00 PM |

if

| Page 16: [29] Deleted | Revision | 1/1/18 1:55:00 PM |

if

| Page 16: [30] Deleted | Revision | 1/1/18 1:55:00 PM |

4a

| Page 16: [30] Deleted | Revision | 1/1/18 1:55:00 PM |

4a

| Page 16: [31] Deleted | Revision | 1/1/18 1:55:00 PM |

start, meaning that the

| Page 16: [31] Deleted | Revision | 1/1/18 1:55:00 PM |

start, meaning that the

| Page 16: [31] Deleted | Revision | 1/1/18 1:55:00 PM |

start, meaning that the

| Page 16: [32] Deleted | Revision | 1/1/18 1:55:00 PM |

in the magnitude

| Page 16: [32] Deleted | Revision | 1/1/18 1:55:00 PM |

in the magnitude

| Page 17: [33] Deleted | Revision | 1/1/18 1:55:00 PM |

exceeds

| Page 17: [33] Deleted | Revision | 1/1/18 1:55:00 PM |

exceeds

| Page 17: [33] Deleted | Revision | 1/1/18 1:55:00 PM |

exceeds

| Page 17: [33] Deleted | Revision | 1/1/18 1:55:00 PM |

exceeds

| Page 17: [34] Deleted | Revision | 1/1/18 1:55:00 PM |

of

| Page 17: [34] Deleted | Revision | 1/1/18 1:55:00 PM |
|---|---|---|

of

| Page 17: [35] Deleted | Revision | 1/1/18 1:55:00 PM |
|---|---|---|

 in Fig. 4b

| Page 17: [35] Deleted | Revision | 1/1/18 1:55:00 PM |
|---|---|---|

 in Fig. 4b

| Page 17: [35] Deleted | Revision | 1/1/18 1:55:00 PM |
|---|---|---|

 in Fig. 4b

| Page 17: [35] Deleted | Revision | 1/1/18 1:55:00 PM |
|---|---|---|

 in Fig. 4b

| Page 17: [35] Deleted | Revision | 1/1/18 1:55:00 PM |
|---|---|---|

 in Fig. 4b

| Page 17: [35] Deleted | Revision | 1/1/18 1:55:00 PM |
|---|---|---|

 in Fig. 4b

| Page 17: [35] Deleted | Revision | 1/1/18 1:55:00 PM |
|---|---|---|

 in Fig. 4b

| Page 17: [35] Deleted | Revision | 1/1/18 1:55:00 PM |
|---|---|---|

 in Fig. 4b

| Page 17: [35] Deleted | Revision | 1/1/18 1:55:00 PM |
|---|---|---|

 in Fig. 4b

| Page 17: [35] Deleted | Revision | 1/1/18 1:55:00 PM |
|---|---|---|

 in Fig. 4b

| Page 17: [35] Deleted | Revision | 1/1/18 1:55:00 PM |
|---|---|---|

 in Fig. 4b

| Page 17: [36] Deleted | Revision | 1/1/18 1:55:00 PM |
|---|---|---|

(i.e., RO)

| Page 17: [36] Deleted | Revision | 1/1/18 1:55:00 PM |
|---|---|---|

(i.e., RO)

| Page 17: [37] Deleted | Revision | 1/1/18 1:55:00 PM |
|---|---|---|

with one another.

| Page 17: [37] Deleted | Revision | 1/1/18 1:55:00 PM |
|---|---|---|

with one another.

| Page 17: [38] Deleted | Revision | 1/1/18 1:55:00 PM |
|---|---|---|

$$\mathbf{H}^{\mathrm{T}}$$

| Page 17: [38] Deleted | Revision | 1/1/18 1:55:00 PM |
|---|---|---|

$$\mathbf{H}^{\mathrm{T}}$$

| Page 17: [39] Deleted | Revision | 1/1/18 1:55:00 PM |
|---|---|---|

those described in Sect. 2.3).

| Page 17: [39] Deleted | Revision | 1/1/18 1:55:00 PM |
|---|---|---|

those described in Sect. 2.3).

| Page 17: [39] Deleted | Revision | 1/1/18 1:55:00 PM |
|---|---|---|

those described in Sect. 2.3).

| Page 17: [40] Deleted | Revision | 1/1/18 1:55:00 PM |
|---|---|---|

the

| Page 17: [40] Deleted | Revision | 1/1/18 1:55:00 PM |
|---|---|---|

the

| Page 17: [40] Deleted | Revision | 1/1/18 1:55:00 PM |
|---|---|---|

the

| Page 17: [40] Deleted | Revision | 1/1/18 1:55:00 PM |
|---|---|---|

the

| Page 17: [40] Deleted | Revision | 1/1/18 1:55:00 PM |
|---|---|---|

the

| Page 17: [40] Deleted | Revision | 1/1/18 1:55:00 PM |
|---|---|---|

the

| Page 17: [40] Deleted | Revision | 1/1/18 1:55:00 PM |
|---|---|---|

the

| Page 17: [40] Deleted | Revision | 1/1/18 1:55:00 PM |
|---|---|---|

the

| Page 17: [40] Deleted | Revision | 1/1/18 1:55:00 PM |
|---|---|---|

the

| Page 18: [41] Formatted | Revision | 1/1/18 1:55:00 PM |
|---|---|---|

Automatically adjust right indent when grid is defined, Widow/Orphan control, Adjust space between Latin and Asian text, Adjust space between Asian text and numbers

**Page 18: [42] Deleted**           **Revision**           **1/1/18 1:55:00 PM**

[revised manuscript text omitted]

Font:Not Bold

| Page 32: [61] Formatted | Revision | 1/1/18 1:55:00 PM |

Font:Not Bold

| Page 32: [62] Deleted | Revision | 1/1/18 1:55:00 PM |

[Figure]

[Figure]

| Page 33: [63] Formatted | Revision | 1/1/18 1:55:00 PM |
|---|---|---|

Font:Not Bold

| Page 33: [63] Formatted | Revision | 1/1/18 1:55:00 PM |
|---|---|---|

Font:Not Bold

| Page 33: [64] Formatted | Revision | 1/1/18 1:55:00 PM |
|---|---|---|

Font:Not Bold

| Page 33: [64] Formatted | Revision | 1/1/18 1:55:00 PM |
|---|---|---|

Font:Not Bold

| Page 33: [64] Formatted | Revision | 1/1/18 1:55:00 PM |
|---|---|---|

Font:Not Bold

| Page 33: [65] Formatted | Revision | 1/1/18 1:55:00 PM |
|---|---|---|

Font:Not Bold

| Page 33: [65] Formatted | Revision | 1/1/18 1:55:00 PM |
|---|---|---|

Font:Not Bold

| Page 33: [65] Formatted | Revision | 1/1/18 1:55:00 PM |
|---|---|---|

Font:Not Bold

| Page 34: [66] Formatted | Revision | 1/1/18 1:55:00 PM |
|---|---|---|

Font:Not Bold

| Page 34: [66] Formatted | Revision | 1/1/18 1:55:00 PM |
|---|---|---|

Font:Not Bold

| Page 34: [67] Formatted | Revision | 1/1/18 1:55:00 PM |
|---|---|---|

Font:Not Bold

| Page 34: [68] Formatted | Revision | 1/1/18 1:55:00 PM |
|---|---|---|

Font:Not Bold

| **Page 34: [69] Formatted** | **Revision** | **1/1/18 1:55:00 PM** |
|---|---|---|

Font:Not Bold

| **Page 34: [70] Formatted** | **Revision** | **1/1/18 1:55:00 PM** |
|---|---|---|

Font:Not Bold

| **Page 34: [71] Formatted** | **Revision** | **1/1/18 1:55:00 PM** |
|---|---|---|

Font:Not Bold

| **Page 34: [72] Formatted** | **Revision** | **1/1/18 1:55:00 PM** |
|---|---|---|

Font:Not Bold

| **Page 34: [73] Formatted** | **Revision** | **1/1/18 1:55:00 PM** |
|---|---|---|

Font:Not Bold

| **Page 34: [74] Formatted** | **Revision** | **1/1/18 1:55:00 PM** |
|---|---|---|

Font:Not Bold

| **Page 34: [75] Formatted** | **Revision** | **1/1/18 1:55:00 PM** |
|---|---|---|

Font:Not Bold

| **Page 34: [76] Formatted** | **Revision** | **1/1/18 1:55:00 PM** |
|---|---|---|

Font:Not Bold

| **Page 34: [77] Formatted** | **Revision** | **1/1/18 1:55:00 PM** |
|---|---|---|

Font:Not Bold

| **Page 34: [78] Formatted** | **Revision** | **1/1/18 1:55:00 PM** |
|---|---|---|

Font:Not Bold

| **Page 34: [79] Formatted** | **Revision** | **1/1/18 1:55:00 PM** |
|---|---|---|

Font:Not Bold

| **Page 34: [80] Formatted** | **Revision** | **1/1/18 1:55:00 PM** |
|---|---|---|

Font:Not Bold

| **Page 34: [80] Formatted** | **Revision** | **1/1/18 1:55:00 PM** |
|---|---|---|

Font:Not Bold

| **Page 34: [81] Formatted** | **Revision** | **1/1/18 1:55:00 PM** |
|---|---|---|

Font:Not Bold

| **Page 34: [82] Formatted** | **Revision** | **1/1/18 1:55:00 PM** |
|---|---|---|

Font:Not Bold

| **Page 34: [83] Formatted** | **Revision** | **1/1/18 1:55:00 PM** |
|---|---|---|

Font:Not Bold

| Page 34: [84] Formatted | Revision | 1/1/18 1:55:00 PM |
|---|---|---|

Font:Not Bold

| Page 34: [85] Formatted | Revision | 1/1/18 1:55:00 PM |
|---|---|---|

Font:Not Bold

| Page 34: [86] Formatted | Revision | 1/1/18 1:55:00 PM |
|---|---|---|

Font:Not Bold

| Page 34: [87] Formatted | Revision | 1/1/18 1:55:00 PM |
|---|---|---|

Font:Not Bold

| Page 34: [88] Formatted | Revision | 1/1/18 1:55:00 PM |
|---|---|---|

Font:Not Bold

| Page 34: [89] Formatted | Revision | 1/1/18 1:55:00 PM |
|---|---|---|

Font:Not Bold

| Page 34: [90] Formatted | Revision | 1/1/18 1:55:00 PM |
|---|---|---|

Font:Not Bold

| Page 34: [91] Formatted | Revision | 1/1/18 1:55:00 PM |
|---|---|---|

Font:Not Bold

| Page 34: [92] Formatted | Revision | 1/1/18 1:55:00 PM |
|---|---|---|

Font:Not Bold

| Page 34: [93] Deleted | Revision | 1/1/18 1:55:00 PM |
|---|---|---|

**(black solid) everywhere. Both reside in the impact parameter coordinate. The solution (red dashed) and true (red solid) refractivity**

| Page 34: [94] Formatted | Revision | 1/1/18 1:55:00 PM |
|---|---|---|

Font:Not Bold

| Page 34: [95] Formatted | Revision | 1/1/18 1:55:00 PM |
|---|---|---|

Font:Not Bold

| Page 34: [96] Formatted | Revision | 1/1/18 1:55:00 PM |
|---|---|---|

Font:Not Bold

| Page 34: [97] Formatted | Revision | 1/1/18 1:55:00 PM |
|---|---|---|

Font:Not Bold

| Page 34: [97] Formatted | Revision | 1/1/18 1:55:00 PM |
|---|---|---|

Font:Not Bold

| Page 34: [98] Formatted | Revision | 1/1/18 1:55:00 PM |
|---|---|---|

Font:Not Bold

| **Page 34: [98] Formatted** | **Revision** | **1/1/18 1:55:00 PM** |
|---|---|---|

Font:Not Bold

| **Page 34: [99] Formatted** | **Revision** | **1/1/18 1:55:00 PM** |
|---|---|---|

Font:Not Bold

| **Page 35: [100] Formatted** | **Revision** | **1/1/18 1:55:00 PM** |
|---|---|---|

Font:Not Bold

| **Page 35: [101] Formatted** | **Revision** | **1/1/18 1:55:00 PM** |
|---|---|---|

Font:Not Bold

| **Page 35: [102] Formatted** | **Revision** | **1/1/18 1:55:00 PM** |
|---|---|---|

Font:Not Bold

| **Page 35: [103] Formatted** | **Revision** | **1/1/18 1:55:00 PM** |
|---|---|---|

Font:Not Bold

| **Page 35: [104] Formatted** | **Revision** | **1/1/18 1:55:00 PM** |
|---|---|---|

Font:Not Bold

| **Page 35: [105] Formatted** | **Revision** | **1/1/18 1:55:00 PM** |
|---|---|---|

Font:Not Bold

| **Page 35: [106] Formatted** | **Revision** | **1/1/18 1:55:00 PM** |
|---|---|---|

Font:Not Bold

| **Page 35: [107] Formatted** | **Revision** | **1/1/18 1:55:00 PM** |
|---|---|---|

Font:Not Bold

| **Page 35: [107] Formatted** | **Revision** | **1/1/18 1:55:00 PM** |
|---|---|---|

Font:Not Bold

| **Page 35: [108] Formatted** | **Revision** | **1/1/18 1:55:00 PM** |
|---|---|---|

Font:Not Bold

| **Page 35: [109] Formatted** | **Revision** | **1/1/18 1:55:00 PM** |
|---|---|---|

Font:Not Bold

| **Page 35: [110] Formatted** | **Revision** | **1/1/18 1:55:00 PM** |
|---|---|---|

Font:Not Bold

| **Page 35: [111] Formatted** | **Revision** | **1/1/18 1:55:00 PM** |
|---|---|---|

Font:Not Bold

| **Page 35: [112] Formatted** | **Revision** | **1/1/18 1:55:00 PM** |
|---|---|---|

Font:Not Bold

| Page 35: [113] Formatted | Revision | 1/1/18 1:55:00 PM |
|---|---|---|

Font:Not Bold

| Page 35: [113] Formatted | Revision | 1/1/18 1:55:00 PM |
|---|---|---|

Font:Not Bold

| Page 35: [114] Formatted | Revision | 1/1/18 1:55:00 PM |
|---|---|---|

Font:Not Bold

| Page 35: [115] Formatted | Revision | 1/1/18 1:55:00 PM |
|---|---|---|

Font:Not Bold

| Page 35: [115] Formatted | Revision | 1/1/18 1:55:00 PM |
|---|---|---|

Font:Not Bold

| Page 35: [116] Formatted | Revision | 1/1/18 1:55:00 PM |
|---|---|---|

Font:Not Bold

| Page 35: [117] Formatted | Revision | 1/1/18 1:55:00 PM |
|---|---|---|

Font:Not Bold

| Page 35: [117] Formatted | Revision | 1/1/18 1:55:00 PM |
|---|---|---|

Font:Not Bold

| Page 35: [118] Formatted | Revision | 1/1/18 1:55:00 PM |
|---|---|---|

Font:Not Bold

| Page 35: [119] Formatted | Revision | 1/1/18 1:55:00 PM |
|---|---|---|

Font:Not Bold

| Page 35: [120] Formatted | Revision | 1/1/18 1:55:00 PM |
|---|---|---|

Font:Not Bold

| Page 35: [121] Formatted | Revision | 1/1/18 1:55:00 PM |
|---|---|---|

Font:Not Bold

| Page 36: [122] Formatted | Revision | 1/1/18 1:55:00 PM |
|---|---|---|

Font:Not Bold

| Page 36: [122] Formatted | Revision | 1/1/18 1:55:00 PM |
|---|---|---|

Font:Not Bold

| Page 36: [123] Formatted | Revision | 1/1/18 1:55:00 PM |
|---|---|---|

Font:Not Bold

| Page 36: [123] Formatted | Revision | 1/1/18 1:55:00 PM |
|---|---|---|

Font:Not Bold